# A Signal Pattern Extraction Method Useful for Monitoring the Condition of Actuated Mechanical Systems Operating in Steady State Regimes

**DOI:** 10.3390/s25041119

**Published:** 2025-02-12

**Authors:** Adriana Munteanu, Mihaita Horodinca, Neculai-Eduard Bumbu, Catalin Gabriel Dumitras, Dragos-Florin Chitariu, Constantin-Gheorghe Mihai, Mohammed Khdair, Lucian Oancea

**Affiliations:** Faculty of Machines Manufacturing and Industrial Management, “Gheorghe Asachi” Technical University of Iasi, 700050 Iasi, Romania; adriana.munteanu@academic.tuiasi.ro (A.M.); neculai-eduard.bumbu@academic.tuiasi.ro (N.-E.B.); catalin-gabriel.dumitras@academic.tuiasi.ro (C.G.D.); dragos-florin.chitariu@academic.tuiasi.ro (D.-F.C.); constantin-gheorghe.mihai@student.tuiasi.ro (C.-G.M.); mohammed.khdair@student.tuiasi.ro (M.K.); lucian.oancea@student.tuiasi.ro (L.O.)

**Keywords:** sensors, state signals, mechanical system, gearbox, signal pattern recognition, condition monitoring

## Abstract

The aim of this paper is to present an approach to condition monitoring of an actuated mechanical system operating in a steady-state regime. The state signals generated by the sensors placed on the mechanical system (a lathe headstock gearbox) operating in a steady-state regime contain a sum of periodic components, sometimes mixed with a small amount of noise. It is assumed that the state of a rotating part placed inside a mechanical system can be characterized by the shape of a periodic component within the state signal. This paper proposes a method to find the time domain description for the significant periodic components within these state signals, as patterns, based on the arithmetic averaging of signal samples selected at constant time regular intervals. This averaging has the same effect as a numerical filter with multiple narrow pass bands. The availability of this method for condition monitoring has been fully demonstrated experimentally. It has been applied to three different state signals: the active electrical power absorbed by an asynchronous AC electric motor driving a lathe headstock gearbox, the vibration of this gearbox, and the instantaneous angular speed of the output spindle. The paper presents some relevant patterns describing the behavior of different rotating parts within this gearbox, extracted from these state signals.

## 1. Introduction

A reasonable assumption in monitoring and diagnostics is that an actuated mechanical system containing periodically rotating mechanical components (MCs) operating in a stationary regime generates state signals to which all MCs contribute additively depending on their state. A state signal can describe vibration, mechanical power, force, torque, angular speed, etc. These state signals contain constant or slowly varying signal components (CSC) and periodically varying signal components (PVSC), where each CSC and PVSC is usually generated by an MC. The description of the health condition of the MC can obviously be conducted by analyzing the time-domain representation of the CSC and PVSC generated by MC. Since the mechanical system naturally mixes all these components into a single state signal, the first problem to solve for monitoring purposes is to separate the CSC and PVSC components generated by each MC from this signal. Of course, at the current state of the art, the CSC components cannot be separated because they have no characteristic elements that link them to the MC. It is obvious that any PVSC generated in the steady-state regime can be approximated as a sum of harmonically correlated sinusoids (with a fundamental and several harmonics). The separation of any PVSC from a state signal is possible because there is a certain feature that links it to the MC during a steady state regime of operation: the frequency of the fundamental component of the PVSC is equal to the rotational frequency of the MC.

Separating a PVSC is not always easy. For various reasons, the three characteristics of the fundamental and harmonic components within a PVSC as sine waves (amplitude, frequency, and phase at the origin of time) may be constant, slightly variable, or severely variable (especially with respect to amplitude). In some situations, the PVSC may be generated only temporarily, for short periods of time. For each situation, synthetic or analytical techniques and methods for a partial or a complete description of the PVSC exist in the literature.

The best-known technique is the Fast Fourier Transform (FFT) of the state signal, which is used in situations where frequencies and amplitudes are constant or vary very little [1,2], and the result is provided as a spectrum. A conversion is made from the time domain to the frequency domain. Each peak in the spectrum describes the average values of the amplitude and frequency of a component in the PVSC (a fundamental or one of its harmonics). All components of all PVSCs are described in the FFT spectrum. Separation of the spectral peaks describing a particular PVSC generated by a particular MC must be conducted using other methods. The FFT technique has been applied in various situations for monitoring and diagnosis of rotating machines [3,4,5,6,7].

When the PVSC characteristics vary greatly with time or especially when the PVSC is generated temporarily, an alternative to the FFT, the wavelet transform, can be used [8,9,10,11,12]. It is systematically applied in various fields, in particular to analyze state signals for monitoring purposes of rotary machines [4,6,7,11,13,14].

A natural question about the steady-state regimes is: what exactly does this separation of a particular PVSC from a status signal mean? It just means that we should somehow obtain the time-domain representation of just this PVSC. Since the period *T* of the fundamental sine wave in the PVSC is known, the simplest way to obtain this representation is to filter the variable part of the status signal (converted to numerical format) with a tunable numerical multiple narrow band pass filter [15] whose pass frequencies are harmonically correlated at 1/*T* Hz, 2/*T* Hz, 3/*T* Hz, and so on (up to the Nyquist limit), while preserving the original phases at the origin of time. The time-domain representation of the desired PVSC appears at the filter output. Indirectly, this is the technique used in this paper. This approach (as well as the FFT) is only correct if any two or more harmonics of fundamentals of different PVSCs do not have the same frequency.

For comparison and analysis purposes, it is easier to consider a time-domain representation of a PVSC (as a periodic signal) replaced with a PVSC described by a single period obtained by appropriate averaging of the PVSC. This approach also covers the situation where the PVSC contains slightly varying sinusoidal components (caused, for example, by a slight variation of the MC speed). This is a brief conceptual description of the approach in this paper.

Finding periodic patterns in state signals (especially vibrations) has been the subject of numerous previous theoretical and experimental approaches in signal processing techniques, particularly in the field of fault detection in rotating machinery. The inverse Fourier transform applied to the logarithm of the magnitudes of the power spectrum obtained by the direct Fourier transform of the state signal, known as the Cepstrum technique [16,17], or some other techniques derived from it have been applied in [12,18,19,20]. Spectral correlation density [21], which examines the correlation between different components of a signal that are related to each other by frequency, or techniques derived from it, was applied in [22,23,24,25]. Some other appropriate techniques are also available: cyclostationary analysis [26,27], autocorrelation [28,29], and averaged cyclic periodogram [30].

The concept of signal pattern recognition, as a relatively new approach to determining patterns within the evolution of state signals, is a well-covered theoretical and experimental research topic [31,32]. Signal pattern recognition is a field of machine learning [33,34] that focuses on defining appropriate algorithms for automatically finding patterns in signals, including deep learning based on neural networks [35]. It is associated with computer-based signal processing, achieved through the use of artificial intelligence. Signal pattern recognition is often used to detect abnormal working conditions, usually to detect mechanical faults. Some interesting achievements in mechanical fault detection on rotary machines are described in refs. [36,37] (rolling bearings condition), refs. [38,39,40] (gears condition), ref. [41] (rotors vibrations), refs. [12,42,43] (induction motors condition), ref. [44] (driving belts condition, a topic also covered in our paper), and refs. [45,46,47] (detection of chatter in cutting processes).

Generally, vibration description signals are analyzed, but other resources are also used, such as force description signals [48], instantaneous angular speed [47], active electric power [49], and electric current [50].

This diversity of approaches and achievements does not limit the perspectives of possible new contributions. There are still enough new accessible resources that can be highlighted and exploited in the identification of PVSC patterns useful for off-line monitoring and predictive diagnostics of rotating mechanical systems operating in steady-state regimes. This is the objective of the present work, which proposes a simple method to extract the PVSC patterns through a selective averaging process of the samples of the state signal. The availability of this method is proved by PVSC extraction from various state signals describing the operation of a lathe gearbox running in a steady-state regime as active electrical power, vibrations, and instantaneous angular velocity. A setup and a theoretical approach already presented and described in [3] will be extensively used here for experimental purposes.

The rest of the paper is organized as follows:-Section 2 presents the materials and methods, mainly the averaging method of state signal samples used to define the PVSC patterns and the experimental setup;-Section 3 presents some experimental results (with the PVSC patterns detected in active electrical power, vibration, and instantaneous angular velocity);-Section 4 is reserved for discussion, with a brief review of the requirements of the averaging method, of the performances in pattern detection, with a summary of the advantages and shortcomings, and future research directions.

## 2. Materials and Methods

### 2.1. The Pattern Extraction Method

An analog signal *s*(*t*) provided by an appropriate sensor placed on a mechanical system running in a steady-state regime is usually described (after analog-to-digital conversion) by a sequence of *p* equidistant numerical samples (with a constant sampling time Δ*t* between any two successive samples), with the *k-*th sample written as *s*[*k*], taken at the time *t* = *k·*Δ*t*.

Suppose the signal *s* contains a PVSC with constant period *T* (or with 1/*T* constant frequency) and at least *m* periods. A signal pattern extraction method of PVSC is proposed below. A pattern of this PVSC, as a time-domain representation over a period *T*, can be obtained mathematically with a good approximation using an extraction method by averaging selected samples (EMASS) at regular time intervals from the signal *s*. This regular time interval is exactly the period *T*. With *m* big enough, the selection rule of these samples results from the definition of a sample *s_T_*[*h*] of this pattern, according to:(1)$$s_{T}\left[h\right]\approx \frac{1}{m}\sum_{i=1}^{m}s\left[h+\left\lfloor \left(i-1\right)\cdot \left.\frac{T}{∆t}\right\rceil \right.\right]\;\;\;\;\;\;\;\;with\;\;h=1,\;2,\ldots ,\left\lfloor \left.\frac{T}{∆t}\right\rceil \right.\;\;\;$$

Here $\left\lfloor \left.x\right\rceil \right.$ is the nearest integer to *x*. If the ratio *n* = *T/*Δ*t* is exactly an integer and the signal *s* has at least *p* = *m·n* samples, then a sample *s_T_*[*h*] of this pattern is more easily written as:(2)$$s_{T}\left[h\right]\approx \frac{1}{m}\sum_{i=1}^{m}s\left[h+\left(i-1\right)\cdot n\right]\;\;\;\;\;\;\;\;with\;\;h=1,\;2,\ldots ,n\;$$

According with Equation (2), a sample *s_T_*[*h*] of this pattern is calculated as the average of *m* equidistant samples from signal *s*, more specifically as the average of these samples: *s*[*h*], *s*[*h* + *n*], *s*[*h + 2·n*], …, *s*[*h* + *m·n*]. Because usually *T*/Δ*t* is not an integer, the description of a sample *s_T_*[*h*] of this pattern with Equation (1) is more exact and always used in the experimental approaches of this work.

If, for simplicity, *p* = *m·n*, then the pattern *s_T_* can be artificially extended to *p* samples so that an extended pattern *s_Te_* of PVSC becomes a signal consisting of the joining of *m* identical periods *s_T_*, one after another. Thus, the extended pattern *s_Te_* can also be defined with *n* concatenated sets of *m* identical defined samples, with $n=\left\lfloor \left.T/\Delta t\right\rceil \right.$. The samples from the *hth* set of extended pattern *s_Te_* (with *h* = 1, 2, …, *n*) are defined with a good approximation as:(3)$$s_{Te}\left[h+\left(d-1\right)\cdot n\right]\approx \;\frac{1}{m}\sum_{i=1}^{m}\;s\left[h+\left\lfloor \left(i-1\right)\cdot \left.\frac{T}{∆t}\right\rceil \right.\right]\;\;\;\;\;\;\;\;with\;d=1,\;2,\;\ldots ,m$$

The left side of Equation (3) describes the concatenation rule of sets. In the extended pattern *s_Te_*, *h* = 1 defines the samples 1, 1 + *n*, 1 + 2*n*, …, 1 + (*m* − 1)*·n*, while *h* = *n* defines the samples *n*, 2*n*, …, *m·n*.

This signal *s_Te_* can be viewed as being generated at the output of a tunable numerical multiple narrow bandpass filter, at whose input the signal *s* is applied (and processed according to Equation (3)). This filter has the pass frequencies on *j*/*T* Hz, with *j* = 1, 2, …, $\left\lfloor \left.2/\Delta t\right\rceil \right.$. Here $\left\lfloor \left.2/\Delta t\right\rceil \right.$ is the Nyquist limit.

As a particular example, the dependence of transmittance by frequency for this filter is partially depicted in Figure 1, for a frequency range between 1 and 300 Hz, as a result of a numerical simulation in Matlab with Δ*t* = 1/50,000 s, *T* = 1/33 s, $n=\left\lfloor \left.50,000/33\;\right\rceil =1515\right.$ and *m* = 60. A numerical sine wave with amplitude of 1 and a frequency *f* in the frequency range is fed into the filter as signal *s*. The amplitude of the output signal *s_Te_* is equal to the transmittance at the frequency *f*.

As clearly indicated in Figure 1, mainly the input signal components (sine waves) having *j*/*T* = *j·33* Hz frequencies pass through the filter unaffected (with undiminished or canceled amplitudes). A zoomed-in detail in area A is depicted in Figure 2 (centered on the pass band frequency of 198 = 6·33 Hz). It is clear that the transmittance is not ideal (because of the small lateral lobes); however, this filter can be used acceptably for the extraction of periodic signal components, as the experimental results will show below.

It is intuitive that this filter does not introduce any phase shift. An illustration of the effectiveness of the proposed EMASS can be realized as follows. The signal *s* is defined to be periodic as a sum of harmonically correlated harmonic components (a fundamental of period *T* and several harmonics with period’s *l·T*) as follows (as an example):(4)$$s\left[k\right]=\sum_{l=1}^{40}l\cdot sin(\frac{2\cdot \pi \cdot l\cdot k\cdot ∆t}{T}+l)\;$$

Here is considered Δ*t* = 1/25,000 s and *T* = 1 s.

The EMASS is used to extract the periodic pattern *s_T_* from the signal *s*, with *m* = 50, according to Equation (2), because $n=T/\Delta t=\left\lfloor \left.T/\Delta t\right\rceil \right.\;$= 25,000. It is expected that the correct operation of the EMASS should be confirmed by the perfect identity between the pattern *s_T_* and the first period of the signal *s*. This identity is confirmed if any difference *s*[*k*] − *s_T_*[*k*] = 0 (also called residual) for any *k* = 1, 2, …, *n*, or for any time *t* expressed as *k·*Δ*t*.

If we plot how these residuals *s*[*k*] − *s_T_*[*k*] evolve over time *k·*Δ*t*, we should obtain a line that coincides with the *t*-axis. Figure 3 shows the time-domain representation of the residuals over a period *T* = 1 s (this being the time duration of the pattern *s_T_*).

Figure 3 proves that this theoretical assumption (with the residuals represented as a line identical to *t*-axis) is not totally true for unknown reasons. However, the maximum peak-to-peak amplitude of the residuals (3 × 10^−10^) is extremely small, so it can be considered that the proposed EMASS and the filter described in Equation (3) work as presumed. This approach also confirms that the EMASS and the filter do not introduce any phase shift.

It is obvious that the best result of extraction and filtering is obtained when $T/\Delta t=\left\lfloor \left.T/\Delta t\right\rceil \right.$. This has already been proved in Figure 3, where we obtained the smallest peak-to-peak residual.

Obviously, the biggest peak-to-peak amplitude of the residual—and the better result for extraction and filtering—is obtained when $\left|T/\Delta t-\left\lfloor \left.T/\Delta t\right\rceil \right.\right|\;$= 0.5. Curve 1 from Figure 4 shows the worst-case time-domain representation of the residuals of the pattern extraction of the signal simulated in Equation (4), with *T* = 1 and Δ*t* = 1/25,000.5 s when $\left|T/\Delta t-\left\lfloor \left.T/\Delta t\right\rceil \;\right.\right|$ = 0.5. There is a 2.49 peak-to-peak amplitude of the residual.

However, this peak-to-peak amplitude of the residual is not significant compared with the peak-to-peak amplitude of *s_T_* (1430.9). By comparison, in Figure 4, curve 2 depicts the time-domain representation of the residuals if Δ*t* = 1/25,000.493 s, and curve 3 depicts the residuals if Δ*t* = 1/25,000.507 s. In both scenarios $\left|T/\Delta t-\left\lfloor \left.T/\Delta t\right\rceil \;\right.\right|\;$= 0.493 as a consequence the peak-to-peak amplitude of the residuals (curves 2 and 3) strongly decreases.

This worst-case $\left|T/\Delta t-\left\lfloor \left.T/\Delta t\right\rceil \;\right.\right|$ = 0.5 often happens because the value of *T*. This worst case can be avoided by resampling the signal *s* (by changing Δ*t*) in order to obtain $\left|T/\Delta t-\left\lfloor \left.T/\Delta t\right\rceil \;\right.\right|$ = 0.

There is another way to illustrate the effectiveness of EMASS. Consider a random noise signal *r_n_* where each sample is generated as a random number in the interval [−5, 5]. Consider that this random noise signal is mixed with a periodical signal *s* (playing the role of a PVSC in this simulation) described as:(5)$$s\left[k\right]=\sum_{l=1,2,20}sin(2\cdot \pi \cdot 25\cdot l\cdot k\cdot ∆t+\frac{\pi }{2})\;\;$$

In Figure 5, curve 1 shows the result of the addition of signals *r_n_ + s* during a period *T* = 1/25 s, where both simulated signals have the same sampling times Δ*t* = 1/100,000 s.

In Figure 5, curve 2 shows the first half of the pattern *s_T_* of this simulated PVSC extracted from the *r_n_ + s* signal using EMASS, with *m* = 50 and *T* = 1/25 s.

Due to the high noise of *r_n_*, the EMASS with *m* = 50 produces a relatively noisy description of the pattern *s_T_*. A better result is shown in Figure 5, curve 3, which shows the second half of a new pattern *s_T_* of this simulated PVSC obtained by EMASS with *m* = 1200.

Curve 4 shows a period *T* of the simulated PVSC. It is obvious that the higher the value of *m*, the better the quality of the PVSC pattern will be. If the PVSC frequency is not strictly constant, this conclusion does not hold. This will be proved experimentally later.

In the previous simulation, it was shown that a very large value of *m* is required to reasonably reduce the influence of noise. This could be seen apparently as an argument against the effectiveness of the EMASS. However, this situation is very rare in practice due to the nature and magnitude of the noise. In our investigations on real state signals, appropriately chosen, the noise level was very low and did not cause any particular problems with respect to the purpose of our work.

With the signal *s* acquired under experimental conditions, as delivered by a sensor placed on an actuated mechanical system operating in the steady-state regime, correct extraction of the pattern *s_T_* of a PVSC by EMASS usually requires the use of Equation (1).

We should mention that it is mandatory to know (or to find somehow) the most accurate value of the period *T* of PVSC. An obvious approach is available for determining the exact value of the period *T*. Knowing an approximate value of *T*, we can set an interval around this value. We systematically change the value of *T* in this interval until we obtain with EMASS (Equation (1)) a pattern *s_T_* with maximum peak-to-peak amplitude. Thus, the correct *s_T_* pattern and the exact value of the period *T* are determined. The correctness of this approach was confirmed experimentally, as will be shown later.

It should also be mentioned that if the signal *s* contains (for example) two periodic components (1, 2) with periods *T*_1_, *T*_2_, and if the *i*-th harmonic of component 1 has the same period as the *j*-th harmonic of component 2 (or *T*_1_*/i* = *T*_2_*/j*), then EMASS will produce distorted results in the description of both patterns *s_T_*_1_ and *s_T_*_2_; these two harmonics will be incorrectly described as belonging to both patterns.

This method of extracting a periodic signal pattern has been partially presented and was the subject of some experimental research related to the PVSC pattern found in instantaneous active electrical power, as a characterization of a three-phase AC asynchronous motor running at idle [51] and related to the PVSC pattern found in the evolution of the roughness of a 2D surface manufactured by milling with a ball nose end mill [52].

### 2.2. The Experimental Setup

An experimental setup already described in detail in [3] is used here (Figure 6 and Figure 7). As an electrically driven mechanical system (with a three-phase AC asynchronous motor), a lathe headstock gearbox is used (with the kinematic scheme partially shown in Figure 7). In this approach, three different state signals provided by appropriate sensors are considered relevant for condition monitoring purposes during a steady-state regime (at idle) of the lathe gearbox: the active electrical power *P_a_* absorbed by the driving motor, the vibration signal vs., and the instantaneous angular speed IAS at the output spindle as signal *I_as_*.

There are some simple reasons for choosing these three state signals. First, it highlights the availability of EMASS for processing different state signals. The best description of the behavior of a mechanical component (part) of a gearbox is in the time domain representation of the mechanical power, which is well reflected in the evolution of the active electrical power. The vibration signal is traditionally used for monitoring and diagnosis. The instantaneous angular speed is strictly related to the evolution of the torque delivered by the electric motor. There are differences and similarities that should be explored and exploited.

The active electric power *P_a_* is mathematically defined [3] based on the signals supplied by a voltage transformer (VT) and a current transformer (CT). The vibration signal is provided by a vibration sensor (VS) [3]. The signal *I_as_* involved in the description of the instantaneous angular speed (IAS) of the spindle is provided by an IAS sensor IASS placed in the jaw chuck of the spindle [3]. All signals are sampled using a numerical oscilloscope [3] connected to a computer. The signal processing was carried out in Matlab R2019b.

The establishment of each signal has been extensively explained in two previous papers (ref. [3] for *P_a_*, and *V_s_*, ref. [53] for *I_as_*). The gearbox is running in steady state at idle (according to the gearing diagram depicted in Figure 7); the MC rotation speeds (experimentally revealed) are highlighted in red.

Next, only the variable parts *P_av_*, vs., and *I_asv_* of these signals are considered experimentally in order to extract by EMASS the patterns of the PVSCs induced by the MCs of the gearbox during the operation at a constant speed of rotation, if these PVSCs are present and have sufficiently high amplitude. These patterns are useful for characterizing the state of each of these MCs (condition monitoring).

## 3. Results

### 3.1. EMASS Applied to Active Electrical Power Signal

Figure 8 shows the time-domain representation of *P_a_* (curve 1), a long sequence of 200 s (with *p_a_* = 5,000,000 samples, Δ*t* = 1/25,000 s, 12-bit resolution). To obtain the *P_av_* signal (as an *s_P_* signal), the constant and the very low frequency variable part of *P_a_* were mathematically subtracted from *P_a_*. This very low-frequency variable part (curve 2) was obtained by low-pass filtering of *P_a_* (using numerical multiple moving average filters [54]). Since this filtering produces false values at the beginning and end of the filtered sequence (zones Z_A_ and Z_B_ in Figure 8), these zones are removed from *P_av_* (so *P_av_* is shorter than *P_a_*, since it has only *p_av_* = 4,984,501 samples).

The time-domain representation of *P_av_* is shown in Figure 9, with a zoomed-in detail (with 1.5 detail of 1.5 s duration from the beginning, shown in the rectangle on the right). This is a first simple proof that *P_av_* is a deterministic signal with a very low level of noise. In the time-domain representation of *P_av_*, many signals are mixed, most of them generated by the MC of the gearbox, as will be shown below.

There is a second, more reliable proof that the *P_av_* signal is deterministic with a low level of noise: how the FFT spectrum of *P_av_* appears, as described in Figure 10.

According to a study performed before [3], each of the labeled peaks (A, B, …, E) indicates the average frequency (inverse of the period) and the average amplitude of the fundamental sine wave of the PVSC generated by the flat belt 1 (A), the flat belt 2 (B), the shaft II (C), the shaft III and the spindle (D), and the shaft I (E). Some other significant peaks describe other sine waves as harmonics of these fundamentals. The frequency (and period as well) of some fundamentals and their harmonics may vary slightly here due to a slight variation in motor rotational speed.

The patterns of each of these PVSCs (as *s_PTA_*, *s_PTB_*, …, *s_PTE_*) can be extracted from the signal *s_P_* by EMASS based on Equation (1). There is a first way to confirm partially the availability of EMASS, related to the average frequencies *f_PA_*, *f_PB_*, …, *f_PE_* of peaks A, B, …, E revealed in Figure 10. For each marked peak in the FFT spectrum (e.g., for the peak A), with EMASS applied at the maximum possible value of *m* (tending to *m_max_* =$\;\left\lfloor p_{av}/(\left.T/\Delta t)-0.5\right\rceil \;\right.$) we search in a small frequency range centered on the value given in the FFT spectrum (e.g., *f_PA_* = 5.34 Hz) for the frequency (period, *T_PA_* = 1*/f_A_*) at which the pattern (*s_PTA_*) has the maximum peak-to-peak amplitude. This is the mean value of the frequency of the respective peak (*f_PA_*), determined indirectly by EMASS, which is expected to be close to that already shown in the FFT spectrum. This hypothesis is fully confirmed with the results depicted in Table 1.

For a more understandable graphical representation, we propose a partial extension of the patterns *s_PTA_*, *s_PTB_*, …, *s_PTE_*, up to the duration of 5 periods of their fundamentals, as *s_PTAe5_*, *s_PTBe5_*, …, *s_PTEe5_*. For example, the *s_PTAe5_* extended pattern is described similarly with Equation (3) as:(6)$$s_{PTAe5}\left[h+\left(d-1\right)\cdot n\right]\approx \;\frac{1}{m}\sum_{i=1}^{m}\;s_{P}\left[h+\left\lfloor \left(i-1\right)\cdot \left.\frac{T_{PA}}{∆t}\right\rceil \right.\right]\;\;\;\;\;\;\;\;with\;d=1,\;2,\;\ldots ,\;5\;$$

These extended patterns *s_PTAe5_*, *s_PTBe5_*, …, *s_PTEe5_* can be obtained by extending the patterns *s_PTA_*, *s_PTB_*, …, *s_PTE_*, each one found by EMASS applied to the *P_av_* signal, with an appropriate value of *m*, with *m < m_max_*.

To show the repeatability of the extended patterns *s_PTAe5_*, *s_PTBe5_*, …, *s_PTEe5_*, there is an interesting possibility: to superimpose two extended patterns of the same PVSC, both calculated with EMASS with the same *m*, the first extended pattern (e.g., as being *s_PTAe5a_*) calculated on the first half of the total number of *P_av_* samples (from 1 to *p_av_/*2, for almost 100 s), and the second extended pattern (e.g., as being *s_PTAe5b_*) calculated on the second half of *P_av_* samples (from *p_av_/*2 to *p_av_*, also for almost 100 s).

Figure 11 shows the superimposed extended patterns *s_PTAe5a_* (curve 1, *m* = 530) and *s_PTAe5b_* (curve 2, *m* = 530) related by the PVSC generated by the flat belt 1. We found that the average frequency *f_PA_* to consider in EMASS for each extended pattern (which produces the maximum peak-to-peak amplitude of the pattern) is slightly different: *f_PAa_* = 5.3368 Hz for *s_PTAe5a_* and *f_PAb_* = 5.3414 Hz for *s_PTAe5b_*. The beginning sample of the second half of the *P_av_* signal (in a first approach *p_av_/*2) from which the second extended pattern (*s_PTAe5b_*) has been deduced is conveniently changed to obtain the most correct possible overlap of the two patterns. A characterization of the flat belt 1 behavior was obtained by means of the extended pattern shape, extracted by EMASS from *P_av_.*

The analytical description of the extended patterns can be found by using the *Curve Fitting Tool* application from Matlab, as the sum of harmonically correlated sinusoidal components (sine waves). Figure 12 shows the *s_PTAe5b_* pattern (as curve 1), the pattern based on the analytical description as the addition of eight harmonically correlated sinusoidal components (as curve 2, an addition of a fundamental F_PA_ and seven harmonics H_PA1_, H_PA2_, …, H_PA6,_ and H_PA8_), and the residuals (curve 3) as the difference between. These sinusoidal components are described in Table 2.

We should mention that a very complicated procedure for finding the shape of the extended pattern for the PVSC generated by the first flat belt has already been presented in previous work [3].

There is another important argument for the correctness of the proposed EMASS. Let us take a sequence of *P_av_* with the duration of 50 periods *T_PA_* (234,300 samples for 9.372 s) from its beginning (as *s_P50_*). Correspondingly, the exact value of the frequency *f_PA_* = 5.33448 Hz and the extended pattern *s_PTAe50_* were determined, with *m* = 50. In Figure 13, the FFT spectrum of the sequence *s_P50_* (in the range 0–40 Hz) is marked as 1. The FFT spectrum of the signal *s_P_*_1*r50*_ resulting from the mathematical extraction of this extended pattern *s_PTAe50_* from the analyzed sequence *s_P50_* is marked as 2. The *k-*th sample of *s_P_*_1*r50*_ is described as *s_P_*_1*r50*_[*k*] = *s_P50_*[*k*] − *s_PTAe50_*[*k*]. It is clear that the fundamental A and its 6 harmonics (A_1_, A2, …, A_6_) have disappeared from the spectrum 2 as a result of the mathematical subtraction of the extended pattern *s_PTAe50_* from the signal *s_P50_*. For any other area (except the blue peaks), the two spectra are identical (spectrum 2 is perfectly superimposed on spectrum 1).

Similarly with Figure 11, Figure 14 shows the superimposed extended patterns *s_PTBe5a_* (curve 1, *m* = 935) and *s_PTBe5b_* (curve 2, *m* = 935) related to the PVSC generated by the flat belt 2. Each extended model is the result of applying EMASS on one half of the signal *P_av_* (or *s_P_* as well). We found that the average frequency *f_PB_* to consider in EMASS for each extended pattern (which produces the maximum peak-to-peak amplitude of the pattern) is quite similar: *f_PBa_* = 9.45223 Hz for *s_PTBe5a_* and *f_PBb_* = 9.45981 Hz for *s_PTBe5b_*.

There are not very significant differences between the extended patterns, except for the peak-to-peak amplitude, which is most likely related to the increase in temperature. It should be noted, however, that the peak-to-peak amplitude of the PVSC of the active electrical power induced by the flat belt 2 is much greater than the PVSC induced by the flat belt 1. Most likely, the flat belt 2 is close to the breakage limit, as it is 35 years older than the first one.

An extended pattern *s_PTBe89_* with 235,494 samples was generated from a sequence of 89 periods *T_PB_* at the beginning of *P_av_*, with *m* = 89 and the best approximation of the average value of the frequency *f_PB_* = 9.446896 Hz in EMASS. This pattern is downsized at first to 234,300 samples and renamed *s_PTBe89*_*. Now the *s_PTBe89*_* extended pattern has the same number of samples as the sequence *s_P50_* and the extended pattern *s_PTAe50_* (both previously used to generate Figure 13). This *s_PTBe89*_* extended pattern can also be mathematically removed from the *s_P_*_1*r50*_ signal; a new signal is obtained as *s_P_*_2*r50*_, with a sample described as *s_P_*_2*r50*_[*k*] = *s_P_*_1*r50*_[*k*] − *s_PTBe89*_*[*k*] = *s_P50_*[*k*] − *s_PTAe50_*[*k*] − *s_PTBe89*_*[*k*]. The downsizing of *s_PTBe89_* until *s_PTBe89*_*[*k*] was necessary to perform the mathematical subtraction above. All three patterns should have the same number of samples.

This signal, *s_P_*_2*r50*_, is described with 234,300 samples taken from the beginning of *P_av_* (as sequence *s_50_*), but after removing from extended signal *s_P50_*, the PVSC generated by the first and second flat belts, through the *s_PTAe50_* and *s_PTBe89*_* extended patterns, is virtual signals. The result of this subtraction is highlighted in the FFT spectrum of signal *s_P_*_2*r50*_ marked with 3 in Figure 15, which is an extension of the result from Figure 13.

It is clear that, in addition to the result and comments of Figure 13, the removal of the PVSC generated by the second flat belt also causes the disappearance of the peaks associated with this component (the fundamental B and the harmonics B_1_ and B_2_) from the FFT spectrum. Obviously, looking at Figure 15, the components B and B_2_ do not disappear completely, but rather some peaks of the FFT spectrum are diminished (the peaks marked with the symbol *). There is a partial explanation for this shortcoming: the periodic component generated by this second flat belt changes its amplitude more strongly in time than the first flat belt, as can be seen in Figure 14, in comparison with Figure 11.

Due to the slight change in rotation frequencies over time, the removal of the extended patterns from a lengthier *P_av_* signal sequence no longer produces the same results in the FFT spectrum, characterized by the complete disappearance of peaks.

Similarly with Figure 11 and Figure 14, Figure 16 shows the superimposed extended patterns *s_PTCe5a_* (curve 1, *m* = 1350) and *s_PTCe5b_* (curve 2, *m* = 1350) related to the periodical component generated by the shaft II. We found that the average frequency *f_PC_* (or period *T_PC_* = 1/*f_PC_*) to consider in EMASS for each extended pattern (which produces the maximum peak-to-peak amplitude of the pattern) is quite similar: *f_PCa_* = 13.75435 Hz for *s_PTCe5a_* and *f_PCb_* = 13.759 Hz for *s_PTCe5b_*. Except for a small difference between peak-to-peak amplitudes, there is a very good coincidence between patterns.

Figure 17 shows the superimposed extended patterns *s_PTDe5a_* (curve 1, *m* = 1720) and *s_PTDe5b_* (curve 2, *m* = 1720) related to the periodical component generated by the shaft III and spindle. We found that the average frequency *f_PD_* (for a period *T_PD_* = 1/*f_PD_*) to consider in EMASS for each extended pattern (which produces the maximum peak-to-peak amplitude of the pattern) is quite similar: *f_PDa_* = 17.37406 Hz for *s_PTDe5a_* and *f_PDb_* = 17.38745 Hz for *s_PTDe5b_*.

Since the shaft III and spindle apparently have the same rotational speeds, the EMASS cannot generate a pattern for each of them. In reality, due to the slippage of belt 2, the rotational speeds are not exactly the same.

Figure 18 shows the superimposed extended patterns *s_PTEe5a_* (curve 1, *m* = 2130) and *s_PTEe5b_* (curve 2, *m* = 2130) related to the periodical component generated by the shaft I. The average frequency *f_PE_* (for a period *T_PE_* = 1/*f_PE_*) to consider in EMASS for each extended pattern (which produces a maximum peak-to-peak amplitude of the pattern) is quite similar: *f_PEa_* = 21.5606 Hz for *s_PTEe5a_* and *f_Peb_* = 21.5781 Hz for *s_PTEe5b_*. A small change in amplitude of the pattern *s_PTEe5b_* is evident.

The mechanical inertia of the gearbox affects the shape and the size of the extended patterns of the PVSC inside *P_av_*. It is obvious that *f_PAa <_ f_PAb_* (Figure 11), *f_PBa <_ f_PBb_* (Figure 14), …, *f_PEa <_ f_PEb_* (Figure 18). This is due to the increase in the motor speed, probably because of the increase in the supply voltage frequency, and certainly because of the decrease in the internal friction due to lubrication.

### 3.2. Some Results Obtained by Analyzing Vibration Signal Using EMASS

A similar analysis can be conducted directly on the vibration signal *V_s_*, written as *s_V_*. This signal was acquired during the same steady-state regime as for active electrical power *P_a_* previously studied, a sequence of 200 s, with *p_a_* = 5 MSa or 5,000,000 samples Δ*t* = 1/25,000 s as the sampling time. This signal is depicted in Figure 19.

This signal vs. describes a beating phenomenon, explained and studied in detail in [55]. The shaft III and the spindle (both with mechanical unbalance) rotate at almost the same instantaneous angular speed; a beating phenomenon (with nodes and anti-nodes) occurs. There is a dominant vibration, shown in an enlarged detail in Figure 19 (on the right), with almost the same frequency as the rotation frequency of the spindle. The partial FFT spectrum of this signal is shown in Figure 20, in a frequency range between 0 and 40 Hz. A zoomed section of this spectrum is shown in the middle, with the same frequency range and the amplitude range severely diminished for magnification: between 0 and 4.2 mV.

It is surprising that in the FFT spectrum of the vs. signal, the same A, B, … E fundamentals of the PVSC are present as previously seen in the FFT spectrum of *P_av_* (Figure 10). This means that the same phenomena are reflected in the time-domain representation of the active electrical power and vibration. Note that the fundamental A is described with diminished amplitude because it has a frequency below the natural frequency of the sensor, 8 Hz. As expected, there is a dominant PVSC with D as fundamental (with a 17.37 Hz frequency and 119.3 mV amplitude) with the origin already explained above in Figure 19.

Even if the vibration description signal contains the beat phenomenon and even if there is a large dominant PVSC (with D as fundamental), the EMASS is able to produce interesting results for monitoring. Related to the behavior of the first flat belt, Figure 21 shows the superimposed extended patterns *s_VTAe5a_* (curve 1, *m* = 533 for almost 2.5 MSa at the beginning of *s_V_*) and *s_VTAe5b_* (curve 2, *m* = 530 for almost the next 2.5 MSa of *s_V_*). Same as above, we found that the right value of average frequency *f_VA_* (for a period *T_VA_* = 1/*f_VA_*) to consider in EMASS for each extended pattern is slightly different: *f_VAa_* = 5.33736 Hz for *s_VTAe5a_* and *f_VAb_* = 5.34151 Hz for *s_VTAe5b_*. As a first approach, since the two patterns appear to be strongly affected by noise, they are plotted in Figure 21 after a numerical low-pass filtering (using a moving average filter with 100 samples in the average). Same as previously overlapped extended patterns, the sample at the beginning of the *s_V_* sequence from which the second extended pattern (*s_VTAe5b_*) has been deduced (almost *p_a_*/2) is conveniently changed to obtain the most correct possible overlap of the two patterns.

As can be seen in Figure 21, there are sufficient similarities between the two extended patterns (despite the relatively long durations between the sequences from which they were derived, almost 100 s) to prove the validity of this resource for describing the condition of the first flat belt using EMASS of vibration signal *s_V_*. It should be noted that in the patterns of this flat belt, the fundamental A has a low amplitude (due to the low sensitivity of the sensor at low frequency), but there are some harmonics with higher amplitudes. It is interesting to highlight the resources offered by the unfiltered time-domain representation of these two overlapped extended patterns, shown in Figure 22, where there are still obvious similarities between them. We propose to realize the extension of the pattern *s_VTAe5b_* to 50 periods, as *s_VTAe50b_*, with 234,000 samples. The partial FFT spectrum of this extended pattern is shown in Figure 23, in a range between 0 and 175 Hz. This figure also shows a zoomed-in detail of this partial spectrum (the same frequency range). This extension of the pattern was necessary to achieve a high-frequency resolution of the FFT spectrum.

The first remark related to Figure 23: the appearance of the FFT spectrum proves that there is no noise in the extended pattern *s_VTAe50b_* (so neither in unfiltered *s_VTAe5a_* nor in *s_VTAe5b_* patterns). In addition, this signal is strictly deterministic, being defined as the sum of strictly harmonically correlated components with frequency spacing exactly equal to the value of *f_VAb_*.

This is another strong argument in favor of the usefulness of EMASS for describing the state of a mechanical component of the mechanical system under investigation. The spectral content highlighted here in the case of the first belt vibration extended pattern was not observed in the case of the active electrical power extended pattern because this power is defined by numerical low-pass filtering of the instantaneous electrical power [3]. The high-frequency harmonics are eliminated by filtering.

It is important to note that all the extended patterns of PVSC from active electrical power presented above can be similarly characterized by means of the FFT spectrum. Both variants of any extended pattern (filtered and unfiltered) can be used to monitor the condition of a rotating MC. The filtered version has the advantage of a quick estimation of the condition.

Related to the behavior of the second flat belt, Figure 24 shows the superimposed extended patterns *s_VTBe5a_* (curve 1, *m* = 935 for almost 2.5 MSa at the beginning of *s_V_*) and *s_VTBe5b_* (curve 2, *m* = 935 for almost the next 2.5 MSa of *s_V_*). The average frequency *f_VB_* (for a period *T_VB_* = 1/*f_VB_*) to consider in EMASS for each extended pattern (which produces a maximum peak-to-peak amplitude of the pattern) is again slightly different: *f_VBa_* = 9.4525 Hz for *s_VTAe5a_* and *f_VBb_* = 9.4596 Hz for *s_VTAe5b_*. Figure 25 shows the same extended patterns but filtered, using a moving average filter with 30 samples in the average.

There are certain similarities between the filtered patterns, but also differences, probably due to the change in temperature of the belt during operation. It should be noted that flat belt 2 introduces a greater peak-to-peak variation in the active electrical power pattern compared to flat belt 1 (see Figure 14 and Figure 11). Conversely, flat belt 2 introduces less variation than flat belt 1 in the description of vibration patterns (see Figure 25 and Figure 21).

Figure 26 shows the superimposed extended patterns generated by the shaft II: *s_VTCe5a_* (curve 1, *m* = 1365 for almost 2.5 MSa at the beginning of *s_V_*, with *f_VCa_* = 13.75399 Hz) and *s_VTCe5b_* (curve 2, *m* = 1365 for almost the next 2.5 MSa samples of *s_V_*, with *f_VCb_* = 13.76565 Hz). Figure 27 shows the same extended patterns treated with a numerical low-pass filtering, using a double-moving average filter (with 37 and 50 samples in the average).

An examination can now be made of the components of one of the unfiltered patterns in Figure 26, e.g., *s_VTCe5b_*. Using the *Curve Fitting Tool* application from Matlab, an approximation of this pattern has been identified based on the first two most representative sinusoidal components. This approximation abeled *s_VTCe5b_*^2*c*^ is described as:(7)$$s_{{VTCe5b}^{2c}}\left[k\right]=3.629\cdot \sin\left(86.57\cdot k\cdot ∆t+2.206\right)+1.324\cdot sin(4152\cdot k\cdot ∆t-1.207)\;$$

Figure 28 shows these two superimposed patterns: *s_VTCe5b_* pattern as curve 2 (already shown in Figure 26) and *s_VTCe5b_*^2*c*^ as curve 3, mathematically described by Equation (7).

A detail from zone Z_A_ illustrating the fit of the two curves is magnified in region Z_B_.

As expected, the angular frequency of the first component, or fundamental (as *ω*_1_ = 86.57 rad/s), is related to the frequency *f_VCb_* (with 2·π·*f_VCb_ ≈ ω*_1_, so 86.4921 rad/s ≈ 86.57 rad/s). The EMASS ensures that the second component of *s_VTCe5b_*^2*c*^ from Equation (7) (with angular frequency *ω*_2_ = 4152 rad/s) is necessarily harmonically correlated with the fundamental. This is totally confirmed because *ω*_2_*/ω*_1_ = 47,9611 ≈ 48. This very small difference is explained by the approximations of the fitting procedure in finding the values of the constants from Equation (7).

Obviously, this second component in Equation (7) must correspond to a vibrational phenomenon. The most plausible explanation for the origin of this phenomenon is that—as shown in Figure 7—a toothed wheel with 48 teeth is mounted on shaft II in free meshing (not transmitting mechanical power). The second component in Equation (7) is generated by the meshing sequence of the teeth of this wheel as a vibration-generating phenomenon with a frequency 48 times higher than the rotational frequency of shaft II.

It is clear that this second component of Equation (7) is always present in the vibration signal *s_V_*. Moreover, it should be emphasized that EMASS is applied to the signal *s_V_*, which is related by a PVSC with period *T_VC_*^48^ = *T_VC_*/48. The extended pattern of this component on five periods *T_VC_*^48^ (as *s_VTC_*^48^*_e5a_* pattern) is shown as curve 1 in Figure 29 (*m* = 3250, on the first 4.94 s of the signal *s_V_*, with the best approximation of the frequency of this PVSC *f_VC_*^48^*_a_* = 660.045986 Hz). The same analysis using EMASS was performed on a new *s_V_* signal sequence starting at the 100th second (*m* = 3250 with the best approximation of the frequency at this variable component *f_VC_*^48^*_b_* = 660.53362 Hz). An extended pattern (*s_VTC_*^48^*_e5b_*) was generated, represented by curve 2 in Figure 29, and shifted appropriately to obtain the best overlap with curve 1. The similarities of these patterns are more than obvious, even though they are described with a small number of samples (190).

Similarly, EMASS can be used to determine if there is a PVSC induced by the 58-toothed gearwheel placed on shaft II as a toothed wheel that transmits mechanical power. The extended pattern of this PVSC on five periods *T_VC_*^58^ (as *s_VTC_*^58^*_e5a_* pattern) is shown as curve 1 in Figure 30 (*m* = 3250, on the first 4.04 s of the signal *s_V_*, with the best approximation of the frequency at this variable component *f_VC_*^58^*_a_* = 797.5646 Hz ≈ 58 · *f_VCa_* = 58·13.75399 Hz = 797.73142 Hz). A new extended pattern of this component on five periods *T_VC_*^58^ (as *s_VTC_*^58^*_e5b_* pattern) is shown as curve 2 in Figure 30 (*m* = 3250, on 4.04 s, starting with the 100th second of the signal *s_V_*, with the best approximation of the frequency at this variable component *f_VC_*^58^*_b_* = 798.1551 Hz ≈ 58 · *f_VCb_* = 58·13.76565 Hz = 798.4077 Hz).

Again, a good similarity of the two patterns from Figure 30 was obtained. Each pattern is described with 155 samples. This proves that the vibrations induced by this toothed wheel occur systematically and that the EMASS is obviously a good option in gearbox condition research. Increasing the sampling rate of the signal *s_V_* increases the number of samples of the patterns from Figure 29 to Figure 30.

As the speed of the electromotor driving the gearbox increases slightly over time (due to the decrease in mechanical load through the effect of the viscosity decreasing of the lubricating oil), all the frequencies of the PVSC studied so far increase slightly over time. Therefore, it should be mentioned that the shape of the patterns generated by the proposed EMASS depends relatively strongly on the value of *m*, especially in the case of high-frequency PVSC (e.g., for the *T_VC_*^48^ periodic component with the patterns already shown in Figure 29 for *m* = 3250). Figure 31 reproduces these two patterns (curves 1 and 2) but also the extended patterns resulting from EMASS with the maximum possible value for *m*, with *m* = 65,700 (curves 3 and 4).

Curve 3 shows the pattern for *m* = 65,700 (as *s_VTC_*^48^*_e5a_*) from the analysis of almost 2.5 MSa from signal *s_V_* (with *f_VC_*^48^*_a_* = 660.19726 Hz). Curve 4 shows the pattern for *m* = 65,700 (as *s_VTC_*^48^*_e5b_*) from the analysis of the next almost 2.5 MSa from signal *s_V_* (with *f_VC_*^48^*_b_* = 660.58252 Hz).

Figure 32 shows the superimposed extended patterns generated by the shaft I, for *m* = 2155, as *s_VTDe5a_* (curve 1, for almost 2.5 MSa at the beginning of *s_V_*, with *f_VDa_* = 21.56087 Hz) and *s_VTDe5b_* (curve 2, for almost the next 2.5 MSa samples of *s_V_*, with *f_VDb_* = 21.57886 Hz). Figure 33 shows the same extended patterns treated with numerical low-pass filtering, using a moving average filter (with 30 samples in the average).

It should be noted that the two filtered extended patterns derived from the vibration analysis by EMASS are more similar for this shaft than for any of the flat belts previously presented (belt 1 in Figure 21 and belt 2 in Figure 25). Similar studies can be performed on the periodic components associated with the rotation of the shaft I and the spindle.

### 3.3. Some Results Obtained by Analyzing Instantaneous Angular Speed Using EMASS

An instantaneous angular speed (IAS) sensor (as IASS) was placed in the jaw chuck of the spindle (Figure 6 and Figure 7). This sensor is actually a stepper motor that plays the role of a two-phase, 50-pole AC generator [53]. At relatively high IAS, this sensor generates two sine-wave signals (equal amplitudes, 90 degrees out of phase) with 50 periods per revolution. These signals are processed appropriately in order to produce IAS, using an interesting approach presented in [56] and appropriately adapted here. This approach is based on determining the angle of rotation of the sensor rotor and the numerical derivative of this angle with respect to time. The time-domain representation of the IAS during an identical steady-state regime considered previously (but shorter, with a duration of only 10 s and a sampling rate of 100,000 s^−1^) is shown in Figure 34.

A detail from zone Z_A_, at the beginning of IAS (with 587 samples, during 5.87 ms), is shown magnified in zone Z_B_. A significant variability of the IAS signal is observed (as peak-to-peak amplitude, with a maximum value of 18.87 rad/s) around the mean value of 109.47 rad/s, corresponding to a mean rotation frequency of 109.47/2/π = 17.4227 Hz. This high variability is due to the fact that this AC generator (actually built to be used as a stepper motor) has mechanical and electrical design imperfections and introduces a PVSC related to these imperfections, which is greatly amplified by numerical derivation.

The frequency of the fundamental sine wave of this PVSC is equal to the rotational frequency (period) of the spindle and shaft III. To use EMASS properly (here with a smaller allowed value for *m*, because the IAS sequence is short), this PVSC should be removed somehow, e.g., using a moving average filter with the number of samples in the average equal to the number of samples per spindle rotation period (here 5739 samples). This means that the pattern generated by the spindle cannot be correctly obtained by EMASS. Figure 35 shows the variable part of the filtered IAS, seen from here onwards as signal *s_I_*.

Of course, all other PVSC of the signal *s_I_* are affected by this filtering, some with reduced amplitude and others being eliminated. However, some of them can still be detected by EMASS.

Figure 36 shows the partial FFT spectrum of the signal *s_I_*, between 0 and 40 Hz, with 0.100715 Hz resolution.

Surprisingly, this spectrum contains the four fundamental sine waves (A, B, C, and E) and some of their harmonics (B_1_, B_2_, C_1_) of the PVSCs previously highlighted in the active electrical power spectrum (Figure 10) and the vibration signal spectrum (Figure 20). This is the first important argument in favor of using the signal *s_I_* in condition monitoring using EMASS. As can be clearly seen, the fundamental sine wave of PVSC generated by the rotation of the spindle and the shaft III in signal *s_I_* (D in Figure 10 and Figure 20) has been completely eliminated from the *s_I_* spectrum from Figure 36, due to the filtering of the IAS signal performed before.

In the same way as above, the EMASS can be used in the signal processing of the signal *s_I_* to extract the extended patterns of the available PVSC.

Related to the behavior of the first flat belt mirrored in signal *s_I_* (with the fundamental A in Figure 36), Figure 37 shows the superimposed extended unfiltered patterns *s_ITAe5a_* (curve 1, *m* = 25 for 467,075 samples at the beginning of *s_I_*) and *s_ITAe5b_* (curve 2, *m* = 25 for the next 467,800 samples of *s_I_*). The average frequency *f_IA_* is slightly different for each pattern: *f_IAa_* = *5.*40021 Hz for *s_ITAe5a_* and *f_IAb_* = 5.4107 Hz for *s_ITAe5b_*.

As expected, there is a good match between the two extended patterns. Surprisingly, the PVSC induced by the flat belt in the signal *s_I_* is detected and described by the IAS sensor even if this sensor is placed far away from this belt.

A detail in zone Z_A_ on Figure 37 is magnified in zone Z_B_. One can see here (especially with respect to the pattern *s_ITAe5a_*) the existence of a variable signal component, the origin of which will be explained later in the discussion of Figure 38 and Figure 39.

Related to the behavior of the second flat belt mirrored in *s_I_* (with the fundamental B in Figure 36), Figure 38 shows the superimposed extended unfiltered patterns *s_ITBe5a_* (curve 1, *m* = 46 for 484,702 samples at the beginning of *s_I_*) and *s_ITBe5b_* (curve 2, *m* = 46 for the next 484,702 samples of *s_I_*). The average value of the frequency used in EMASS was *f_IBa_* = 9.48999 Hz and *f_IBb_* = 9.49061 Hz.

A detail from zone Z_A_ (with 370 samples, during 3.7 ms) is shown magnified in zone Z_B_. This is an example of a distortion phenomenon of the extended patterns already anticipated in Section 2 of this paper.

It should be noted that the PVSC generated by spindle and shaft III in the signal *s_I_* has not been completely removed by the IAS filtering; some of its upper harmonics still remain in the signal *s_I_*. Only the fundamental sine wave D of this PVSC has been completely removed, as shown in Figure 36. It was found that the period of the 369th harmonic of the fundamental of the PVSC generated by the second belt (as *T_IBa_*^369^ = *T_IBa_/*369 or *T_IBb_*^369^ = *T_IBb_/*369) is practically equal to the period of the 201st harmonic of the fundamental of the PVSC generated by the spindle (as *T_IDa_*^201^ = *T_IDa_*/201 or *T_IDb_*^201^ = *T_IDb_*/201), in other words, 369*·f_IBa_ ≈*201*·f_IDa_ ≈* 3501.80 Hz or 369*·f_IBb_ ≈* 201*·f_IDb_ ≈* 3502.035 Hz. For this reason, the sinusoidal component with period *T_IDa_*^201^ (or *T_IDb_*^201^) appears in the extended pattern *s_ITBe5a_* (or *s_ITBe5b_*) as the false sinusoidal component *T_IBa_*^369^ (or *T_IBb_*^369^), as shown in Figure 38 in zone Z_B_. Also in these extended patterns appears any other sinusoidal component having the period *T_IDa_*^2*0*1/^*j* ≈ *T_IBa_*^369^*/j* (or *T_IDb_*^201/^*j* ≈ *T_IBb_*^369^*/j*).

Of course, it is possible to find the extended patterns *s_ITB_*^369^*_e5a_* and *s_ITB_*^369^*_e5b_* of the false variable component with the fundamental with very small period *T_IBa_*^369^ and *T_IBb_*^369^. These extended patterns (with 145 samples) were each determined with EMASS as before, on almost half the number of samples (495,900) from signal *s_I_* (for *m* = 17,100), and are shown in Figure 39.

As can be clearly seen in Figure 39, there are some similarities between these patterns (as shapes, not as amplitudes), already found earlier in Figure 38 as shown in region Z_B_. We discovered that these false signal variable components, *T_IDa_*^201^ and *T_IDb_*^201^, also occur in the *s_ITAe5a_* and *s_ITAe5b_* extended patterns already revealed in Figure 37 (highlighted in the region Z_B_), as having *T_IAa_*^648^ (or *T_IAb_*^648^) as harmonics of the fundamental A within the PVSC generated by the flat belt 1.

Related to the behavior of shaft II mirrored in signal *s_I_* (with fundamental C in Figure 36), Figure 40 shows the superimposed extended unfiltered patterns *s_ITCe5a_* (as curve 1, *m* = 68 on 491,776 samples at the beginning of *s_I_*) and *s_ITCe5b_* (as curve 2, *m* = 68 for the next 491,776 samples of signal *s_I_*). The average value of the frequency used in the EMASS was *f_ICa_* = 13.826942 Hz and *f_ICb_* = 13.79066 Hz. Since there are ten periods on each extended pattern, it is obvious that the amplitude of the first harmonic (described in Figure 36 with the peak C_1_) is much bigger than the fundamental sine wave, described in Figure 36 with the peak C.

Related to the behavior of shaft I mirrored in signal *s_I_* (with fundamental E in Figure 36), Figure 41 shows the superimposed extended unfiltered patterns *s_ITEe5a_* (as curve 1, *m* = 106 for 489,720 samples at the beginning of *s_I_*) and *s_ITEe5b_* (as curve 2, *m* = 106 for the next 489,720 samples of *s_I_*).

The average value of the frequency used in the EMASS was *f_IEa_* = 21.6422 Hz and *f_IEb_* = 21.64315 Hz. In Figure 40 and Figure 41, false variable components also appear, as described earlier. Eliminating these false variable components in the extended patterns requires a simple approach, such as determining their mathematical description (by curve fitting, as was conducted earlier in Figure 12) and removing them from the PVSC.

The EMASS can be used to process state signals produced by many other types of sensors describing a steady-state regime at idle or during a working process.

## 4. Discussion

The facilities of numerical description and sampling of the signal sequences generated by the sensors (such as resolution, sampling rate, and number of samples), as well as the facilities for assisted computation, allow their processing by various relatively simple numerical techniques and methods.

Among these techniques, this paper proposes an extraction method by averaging selected samples (EMASS) at regular time intervals as a procedure for determining the pattern of any periodically varying signal component (PVSC) present in state signals during a steady-state regime of a driven mechanical system, which gives interesting experimental results for three different signals: active electrical power, vibration, and instantaneous angular speed. These state signals were provided by appropriately different sensors placed in different locations on a lathe gearbox headstock. The patterns found in these signals using the proposed EMASS characterize the correct or incorrect functioning of mechanical components (MC) of the mechanical system in a steady-state regime, with a constant speed of rotation, and are useful for offline monitoring of their condition.

### 4.1. A Brief Overview of the Requirements for Pattern Extraction Using EMASS

The computer programs conceived by us in Matlab and only occasionally an application in Matlab (*Curve Fitting Tool*) were used to prepare, apply, and analyze the proposed EMASS. To find the samples of the pattern of any PVSC within the state signal (based on Equation (1)), or the samples of any extended pattern (based on Equation (3)), it is necessary to know:-The samples of the variable part of the state signal are long enough in duration.-The exact value of the sampling time (sampling interval) Δ*t* of the numerical description of the state signal is constant and small enough.-The number of samples of this signal.-The exact value of period *T* (or frequency) of the PVSC.-The number of time intervals is conveniently chosen (*m*), preferably as large as possible.

The most difficult issue in the application of EMASS is the determination of the exact value of the period *T* (or frequency) of the PVSC. In our approach, we have defined and validated an efficient method: in a chosen (small) range for the period (frequency) value, we search for the value at which the peak-to-peak amplitude of the resulting pattern found by EMASS is maximal. The range is centered on the approximate value of the period (frequency) of the PVSC given by the FFT spectrum or by the kinematic scheme of the actuated mechanical system. After each search result, the range is narrowed, and the search is repeated a sufficient number of times to obtain the most accurate period (frequency) value.

The efficiency of this method of accurately determining the value of the PVSC period (frequency), which was systematically used in this paper, can be comparatively demonstrated as shown in Figure 42. In the case of the PVSC of the active electric power induced by the first flat belt (Figure 7), the extended pattern *s_PTAe5a_* was found (Figure 11, as curve 1, for *m* = 530, for which the exact frequency *f_PAa_* = 5.33748 Hz was determined). This extended pattern is redrawn identically as curve 1 here below, in Figure 42.

In Figure 42, curve 2 shows an extended pattern (as *s_PTAe5a−_*) under the same condition but determined by EMASS with an intentionally imprecise, very slightly lower frequency (as *f_PAa-_* = 5.335 Hz), and curve 3 shows this extended pattern (as *s_PTAe5a+_*) but determined by EMASS with an intentionally imprecise, very slightly higher frequency (as *f_PAa+_* = 5.339 Hz). The difference between the extended patterns is obvious (also favored by the large value of *m*), despite the very small changes in frequency. It is obvious that the extended pattern 1, corresponding to the exact frequency, has the largest peak-to-peak amplitude and more accurately describes the behavior of the flat belt 1.

It is important to note that even in the absence of an approximate period (frequency) value of a variable periodic component, it is possible to use the proposed EMASS, starting with a sufficiently large frequency range. The period (frequency) value for a possible PVSC is found to be the period (frequency) that produces a pattern with the largest peak-to-peak amplitude.

When two extended patterns (or extended patterns as well) with obvious similarities that characterize the same variable signal component (at different time instants) are to be compared (by overlapping, e.g., Figure 11, Figure 14, Figure 16, etc.), one of them is taken as the reference, and the other is shifted until the best overlap is obtained. Shifting is accomplished by changing the first sample of the state signal used in the extended pattern determination. Any offset component (or DC bias as well) of the extended pattern should be eliminated.

If the two patterns have relatively large differences in shape, for a correct overlapping, the first sample of the state signal used to find each pattern is shifted so that each pattern starts at the phase origin of its fundamental. This phase origin is converted in zero-crossing time. The description of the phase origin of the fundamental was found using the *Curve Fitting Tool* from Matlab. This technique has been used systematically below to obtain the next figures. A method of curve fitting in Matlab—slower but giving accurate results—has been developed [3] and used by us.

### 4.2. Consideration on the Capability of Patterns to Reflect Changes of Steady-State Regimes

If these patterns are useful for monitoring, it is expected that as the stationary operating conditions of the gearbox change, the behavior of the various MCs will also change, which will be reflected in the change in shape of these patterns. This assumption is briefly confirmed below. A recording of the time-domain representation of the active electric power (the same number of samples and sampling time as before) has been made for a new steady-state regime characterized by the operation of only the electromotor, the flat belt 1, and the shaft I. All the electromagnetic clutches are disengaged; the shafts II, III, and the spindle do not rotate. Figure 43 shows the extended patterns *s_PTAe5a_* (with *m* = 530) generated by the flat belt 1 in the active electrical power during the two steady-state regimes: the extended pattern 1 (from the study above, with all the MC of the gearbox in rotation, a duplication of curve 1 from Figure 11, *f_PAa_* = 5.33748 Hz) and, with overlap, the extended pattern 2 (for this new gearbox configuration, *f_PAa_* = 5.386069 Hz). Since the mechanical power transmitted through the belt is lower in the new gearbox configuration, the shape and peak-to-peak amplitude of the extended pattern 2 are significantly different.

Similarly, Figure 44 shows the extended patterns *s_PTEe5a_* (with *m* = 2130) produced by the shaft I during the same two compared steady-state regimes: curve 1 (a duplication of curve 1 from Figure 18, *f_PEa_* = 21.5606 Hz) and, with overlap, curve 2 (for this new gearbox configuration, *f_PEa_* = 21.7923 Hz).

The peak-to-peak amplitudes for the extended pattern 2 are greatly reduced for the same reason: less mechanical power is being transmitted through this shaft in this new gearbox configuration.

### 4.3. On the Capability of EMASS to Extract Patterns of PVSC of Small Amplitude

This capability has already been fully demonstrated experimentally in the case of the analysis of the signal *s_V_* describing the vibration (the patterns from Figure 29, Figure 30 and Figure 31), but also in the case of the analysis of the signal *s_I_* describing the variable part of the filtered instantaneous angular speed (the patterns from Figure 38 and Figure 39).

It can be shown that the proposed EMASS can also detect patterns of small amplitude PVSC in the *s_P_* signal. Thus, the extended pattern of PVSC generated by the electric motor in the *s_P_* signal could be detected as a time domain representation (*s_PTMe5_*) with the gearbox in the configuration shown in Figure 7. The value of the frequency of the fundamental sine wave within PVSC was sought between 24 and 25 Hz. The EMASS applied to the first half of the signal *s_P_* produced an extended pattern (as *s_PTMe5__a_* with *m* = 2465, *f_PMa_* = 24.69481 Hz) drawn as curve 1 in Figure 45; the second half produced an extended pattern (as *s_PTMe5__b_* with *m* = 2465, *f_PMa_* = 24.712903 Hz), drawn as curve 2 in Figure 45. Each extended pattern starts at a zero-crossing moment of its fundamental.

Figure 46 describes similarly the extended patterns of PVSC generated by the motor during the new configuration of the gearbox (only the motor, the flat belt 1, and the shaft I in rotary motion).

As can be clearly seen, there are relatively good similarities between these extended patterns 1 and 2 drawn in Figure 45 and Figure 46. The differences are probably due to heating during operation (the patterns 1 are generated by analyzing 100 s of the status signal; the patterns 2 are generated by analyzing the next 100 s of the status signal).

### 4.4. On the Capability of the EMASS to Detect Patterns for High-Frequency (Small Period) PVSC

This capability has already been fully demonstrated before in the analysis of the vibration signal *s_V_* (as shown in Figure 28, Figure 29, Figure 30 and Figure 31) and instantaneous angular speed signal *s_I_* (as shown in Figure 37, Figure 38 and Figure 39). Since the active electrical power is defined as a result of low-pass numerical filtering of the instantaneous electrical power, the high-frequency variable components are not available inside the signal *s_P_*.

There is an interesting reason why the vibration description signal is best suited for the pattern-based research of high-frequency periodic variable components: the sensor used is a generator type that provides at its output an electric voltage *s_V_* proportional to the derivative (velocity) of the vibratory motion of the support on which it is placed. This derivative favors the description of high-frequency components (greatly increasing their amplitude). The use of an accelerometer would be even more appropriate since it provides a voltage proportional to the second derivative of the vibratory motion.

For a better utilization of the proposed EMASS in this regard, the conversion rate of the signal being studied (e.g., *s_V_*) must necessarily be greatly increased. Because of the relatively low sampling rate used in our research (25,000 samples per second for *s_P_* and *s_V_* signals), the extended patterns in Figure 30 contain only 38 samples per period.

### 4.5. A Summary of the Benefits of Using EMASS

The main advantage of the proposed EMASS is the obtaining, by numerical calculation, a pattern characterizing the functioning of any rotating mechanical component of the mechanical system operating in the steady-state regime, associated with its rotation period. Compared to a considered reference pattern, its changes over time (due to wear, failure, or change of operating conditions) can be used for offline condition monitoring of the MC.

Characterization patterns of PVSC using EMASS can be determined from signals generated by any type of sensor capable of describing fast phenomena. In our work, we have exemplified signals describing active electrical power, vibration, and instantaneous angular velocity.

Each pattern found by EMASS is, in fact, a sum of sinusoidal components with a fundamental and several harmonics. After being determined by EMASS, each pattern can be extended and partially described with amplitudes and frequencies of its sinusoidal components by FFT analysis (e.g., according to Figure 21), or fully described analytically using the *Curve Fitting Tool* application in Matlab (e.g., according to Figure 12 and Table 2) or any other curve fitting procedure applied to the pattern. The EMASS provides a synthetic description of a pattern (through the coordinates of its points), while a curve-fitting procedure provides an analytical description of this pattern with a formula, as a sum of sinusoidal components, giving the values of amplitude, frequency, and the phase of the time origin of each one. The condition of an MC, or the anomalies in its operation, can be evaluated in two different stages: first, roughly, relatively quickly, based on the shape and peak-to-peak amplitude of its pattern; second, more precisely, based on the analytical description of the fundamental sinusoidal components or/and their harmonics. The evaluation result in each stage can be provided automatically by a computer.

It should be mentioned that obtaining the analytical description of the pattern of an MC can also be conducted directly, by analyzing the state signal with the Curve Fitting Tool application, and not by analyzing the synthetic description of the pattern provided by EMASS, as proposed in this paper. Unfortunately, the direct analysis of the state signal (already proposed in [3]) is very difficult, time-consuming, and subject to possible errors. This is because the operation with the Curve Fitting Tool application usually assumes that the sinusoidal components that could belong to a pattern are automatically searched and selected not by their harmonic frequency correlation but by the value of the amplitudes, descending from the largest to the smallest.

We believe that the monitoring and diagnosis based on the study of EMASS patterns have the advantage of being easy to implement and apply in the automotive industry, in manufacturing systems, and, in general, in any work process supported by mechanical systems. Our research has reached the stage of experimental validation in the laboratory. For the future, we are preparing the practical application and the formulation of some appropriate industrial implementation strategies.

### 4.6. Some Shortcomings of Using the Proposed EMASS

The first major shortcoming in the application of the proposed EMASS is that it cannot automatically identify and eliminate false components within the patterns. Two PVSCs with periods *T*_1_ and *T*_2_ (or with fundamental frequencies of 1/*T*_1_ and 1/*T*_2_) may have interferences of their harmonics, e.g., the *i^st^* harmonic for *T*_1_ and the *j^st^* harmonic for *T*_2_, if *T*_1_*/i* = *T*_2_*/j*. The proposed EMASS finds (incorrectly) the harmonic with period *T*_1_*/i* described in the pattern of the component with period *T*_2_ and (also incorrectly) the harmonic with period *T*_2_*/j* described in the pattern of the component with period *T*_1_. This shortcoming has already been highlighted in the comments to Figure 38. Of course, this shortcoming is also present when other signal component analysis techniques, such as FFT, are in use.

If the period *T* of the VPSC whose pattern is to be extracted is not constant and varies very slightly (as systematically happens in our experiments), then a second major shortcoming arises: the shape and peak-to-peak amplitude of the pattern depend strongly on the value of *m*. The larger *m* is, the smaller this amplitude will be (and the upper harmonics of the pattern will be greatly attenuated). This phenomenon has already been observed in Figure 31 (where the *T_VC_*^48^ plays the role of *T*). In this situation also a third shortcoming occurs: a correct pattern extraction requires the exact determination of the average period (frequency) on each analyzed sequence of the state signal.

Unfortunately, the use of small values of *m* introduces the fourth shortcoming, which is highlighted in Figure 47. It shows the extended pattern *s_PTDe5a_*_1_ of the VPSC obtained by EMASS generated by the shaft III and the spindle in the signal *s_P_* extracted from the first twenty *T_PDa_*_1_ periods (curve 1, *m* = 20, *f_PDa_*_1_ = 17.35793 Hz) and a similar pattern, *s_PTDe5a_*_2,_ extracted from the first 100 *T_PDa_*_2_ periods (curve 2, *m* = 100, *f_DPa_*_2_ = 17.3638 Hz).

A detail from zAA detail from zone Z_A_ is shown in zone Z_B_. In zone Z_B_ an abnormal jump (discontinuity) can be observed on both curves, in zone Z_C_. The smaller the *m* is, the larger the jump will be. This means that the patterns over a period are incorrectly described; the ordinate of the last sample of the current period (here the second in Z_A_) of the extended pattern is very different from the ordinate of the first sample of the next period (here the third in Z_A_). In other words, a period of the extended pattern is incorrectly defined; it does not begin and end with points that have very close, almost identical ordinates. We should note that, in fact, this jump exists in all the patterns presented so far, but because a high value of *m* is practically negligible.

It is obvious that the value of *m* should be as large as possible. It is also obvious that a large value of *m* distorts the shape of the VPSC pattern when its frequency is slightly variable. This disadvantage is partially eliminated if the monitoring and diagnosis are limited to the comparison of two patterns of the same VPSC generated by an MC under the same operating conditions at different times, with the patterns extracted using EAMSS mandatory with *m* having the same value. This was conducted systematically in this study.

As already mentioned, a correct and complete description of the patterns (especially those with a short fundamental period) implies the need to use a sampling rate (frequency) as high as possible in the numerical description of the analyzed state signal. A higher resolution of the numerical signal (in our research, the resolution expressed in bits is 12) is also useless.

### 4.7. Future Research Directions

Firstly, we intend to find a method of improving the accuracy of patterns for the situation when the period of the fundamental of VPSC is slightly variable. In other words, to reduce the negative influence of high values of *m* on the shape of the pattern.

Second, we want to investigate whether numerical resampling of the state signals increases the accuracy of the extracted patterns. In other words, to change the value of the sampling time Δ*t* for each PVSC (by resampling) such that the ratio *T/*Δ*t* is an integer, regardless of the value of *T*.

In a future approach, we aim to extract patterns by EMASS using very high sampling rates of signals, especially in vibration signals (as they are the easiest to acquire). The vibration description signals have several important advantages: they are easy to obtain using simple sensors, they provide signals less affected by attenuation or distortion phenomena (compared to the active electrical power), and they do not introduce significant measurement anomalies (compared to the instantaneous angular speed). We want to obtain highly reliable patterns describing the condition of some other MCs (e.g., bearings, gears, etc.) or even cutting tools during a working process of machine tools (e.g., for milling tools, with patterns that characterize the involvement of each tooth of the tool in the cutting process).

We also intend to highlight the research resources offered by the application EMASS to any state signal containing PVSC and, in particular, to reveal and investigate the patterns in instantaneous electrical power absorbed by the drive motors used to actuate mechanical systems. The description signal of the active electrical power (used until now, coming from the low-pass filtering of the instantaneous electrical power) has some limitations: it strongly attenuates (or even eliminates) the higher harmonics.

We also propose to investigate how to reduce the vulnerability to noise for small peak-to-peak amplitude patterns.

## Figures and Tables

**Figure 1 sensors-25-01119-f001:**
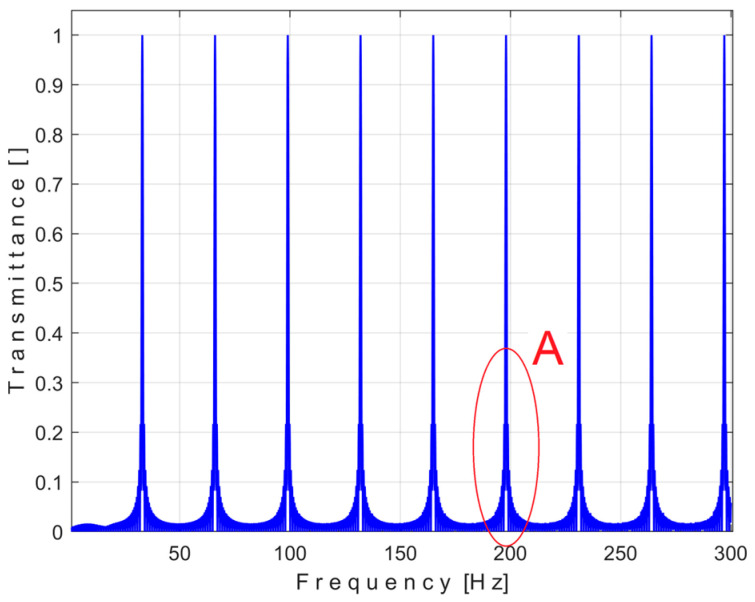
Filter transmittance versus frequency (numerical simulation (Δ*t* = 1/50,000 s, *T* = 1/33 s, $n=\left\lfloor \left.50,000/33\;\right\rceil =1515\right.$ and *m* = 60).

**Figure 2 sensors-25-01119-f002:**
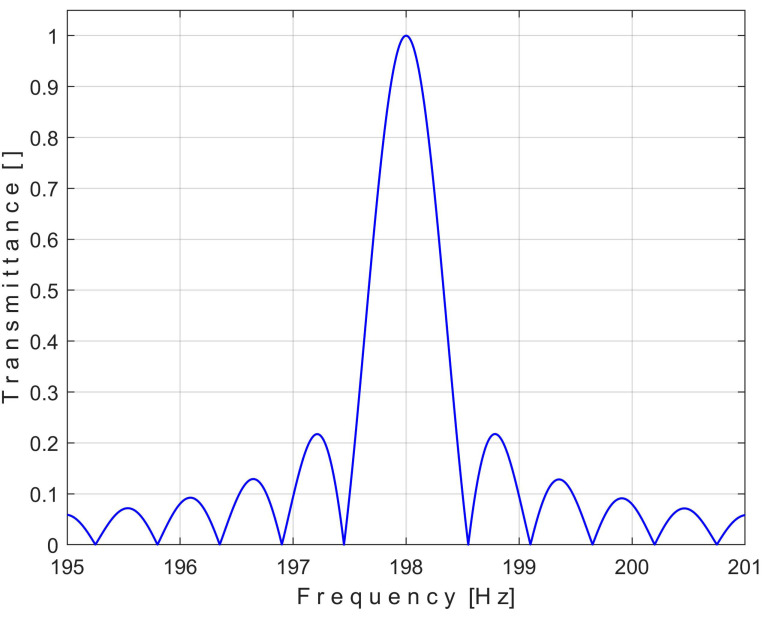
A zooming in in area A from Figure 1.

**Figure 3 sensors-25-01119-f003:**
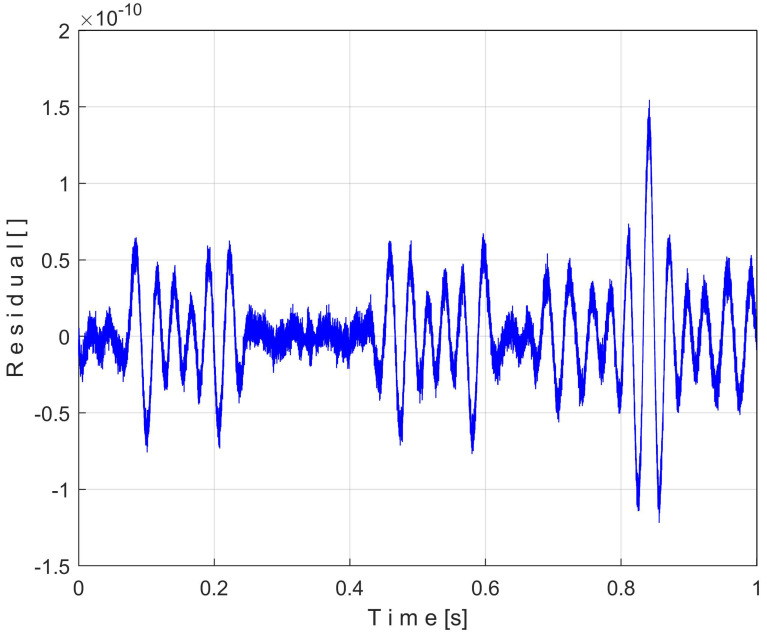
The time-domain representation of the residuals *s*[*k*] − *s_T_*[*k*] over time *k·*Δ*t*, with *k* = 1, 2, …, *n*, *T* = 1 s, Δ*t* = 25,000.

**Figure 4 sensors-25-01119-f004:**
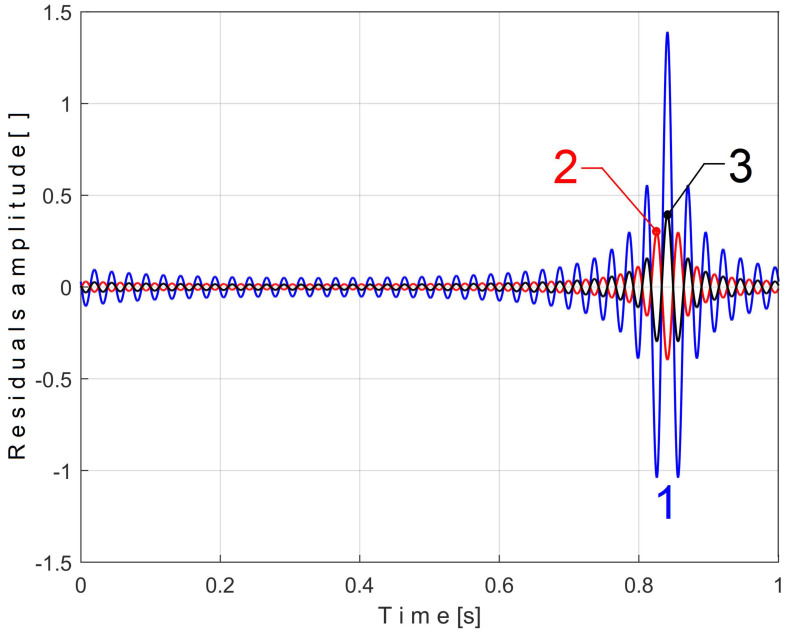
The time-domain representations of the residuals *s*[*k*] − *s_T_*[*k*] over time *k·*Δ*t*, with *k* = 1, 2, …, *n*, *T* = 1 s. Curve 1—Δ*t* = 1/25,000.5 (with $\left|T/\Delta t-\left\lfloor \left.T/\Delta t\right\rceil \right.\right|$ = 0.5); Curve 2—Δ*t* = 1/25,000.493 (with $\left|T/\Delta t-\left\lfloor \left.T/\Delta t\right\rceil \;\right.\right|$ = 0.493); Curve 3—Δ*t* = 1/25,000.507 (with $\left|T/\Delta t-\left\lfloor \left.T/\Delta t\right\rceil \;\right.\right|$= 0.493).

**Figure 5 sensors-25-01119-f005:**
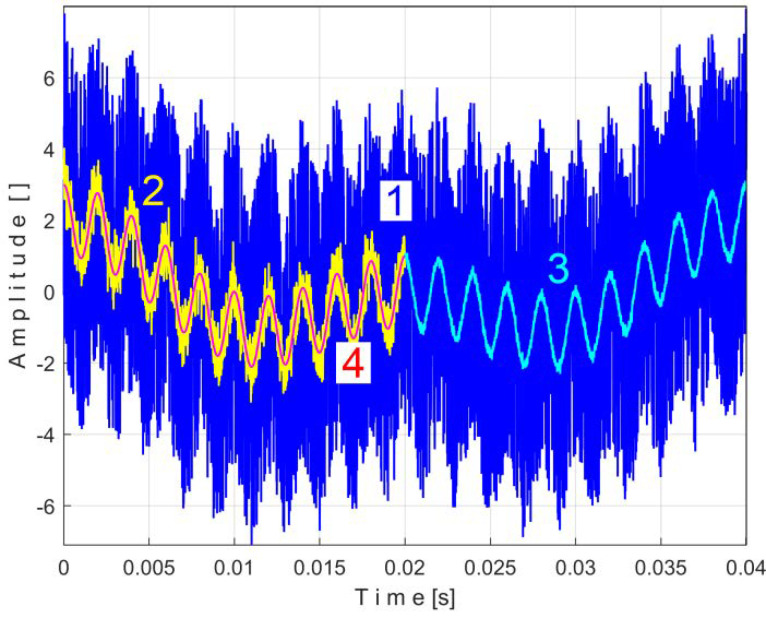
1—A period *T* = 1/25 s of the signal *r_n_ + s*; 2—The first half of the pattern *s_T_* with *m* = 50; 3—The second half of the pattern *s_T_* with *m* = 1200; 4—A period *T* of signal *s*.

**Figure 6 sensors-25-01119-f006:**
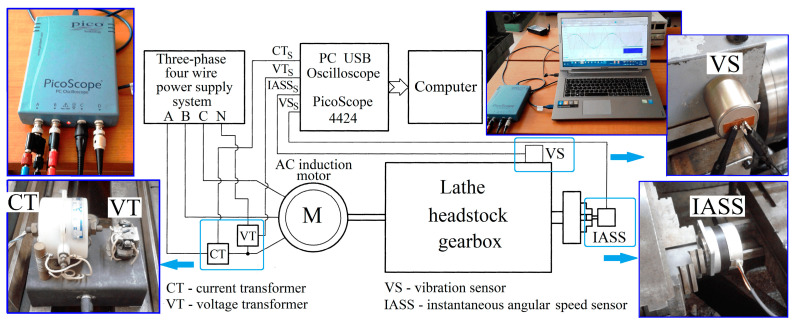
A description of the experimental setup [3].

**Figure 7 sensors-25-01119-f007:**
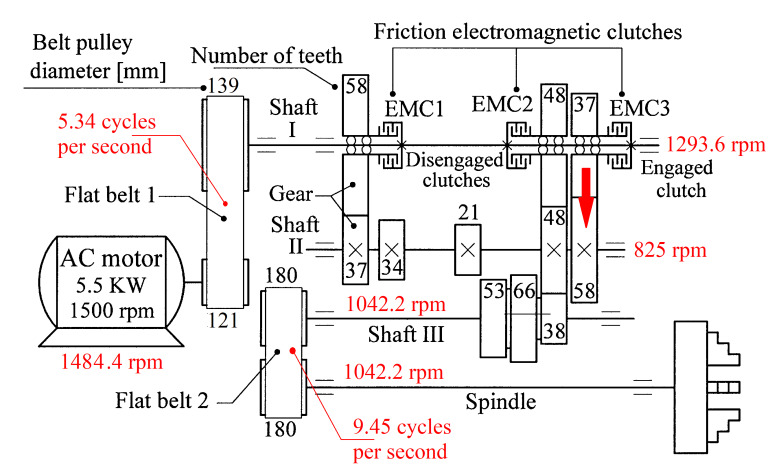
The gearing diagram of the lathe headstock gearbox [3].

**Figure 8 sensors-25-01119-f008:**
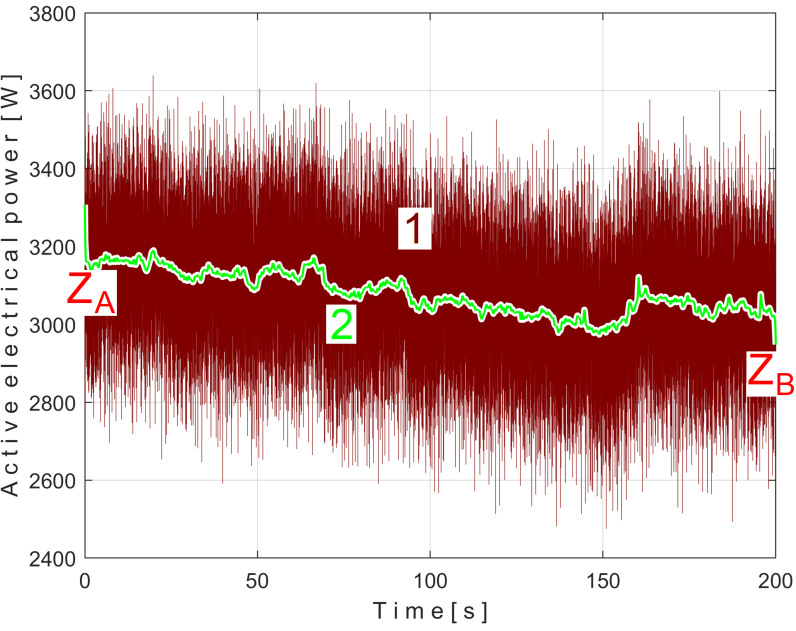
The time-domain representations of: 1—*P_a_*; 2—the very low frequency variable part of *P_a_*.

**Figure 9 sensors-25-01119-f009:**
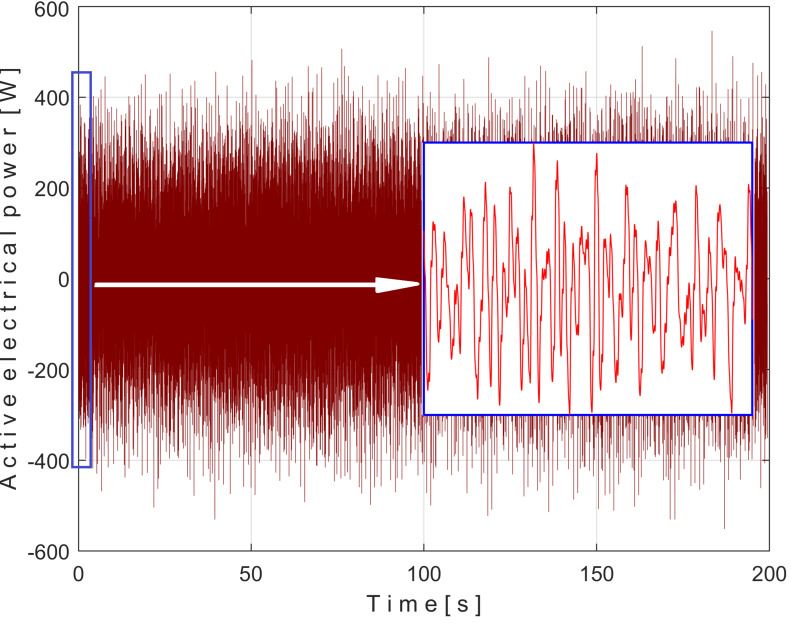
The time-domain representation of *P_av_*.

**Figure 10 sensors-25-01119-f010:**
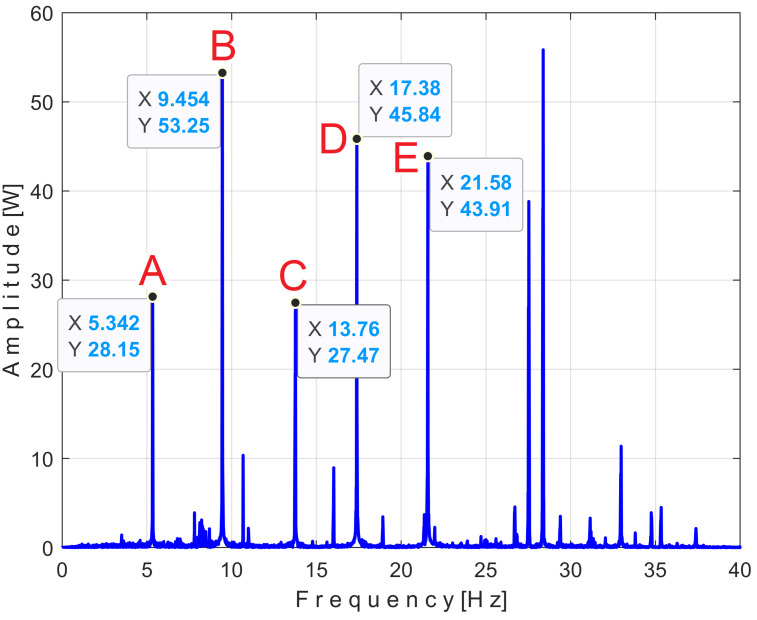
The FFT spectrum of *P_av_* with A, B, …, E the fundamentals of PVSC generated by these MCs: A—the flat belt 1; B—the flat belt 2; C—the shaft II; D—the shaft III and the main spindle; E—the shaft I.

**Figure 11 sensors-25-01119-f011:**
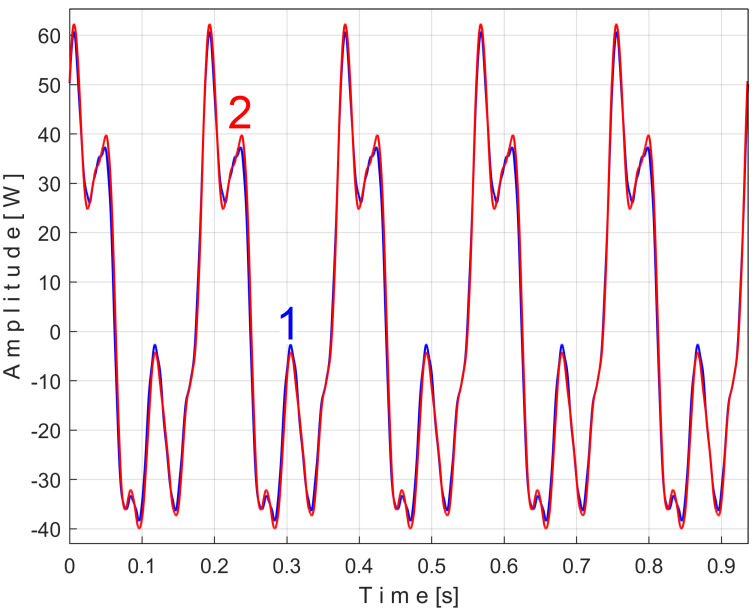
The extended patterns (*m* = 530) describes the PVSC generated by the first flat belt in signal *s_P_*: 1—*s_PTAe5a_* (*f_PAa_* = 5.33748 Hz); 2—*s_PTAe5b_* (*f_PAb_* = 5.34162 Hz).

**Figure 12 sensors-25-01119-f012:**
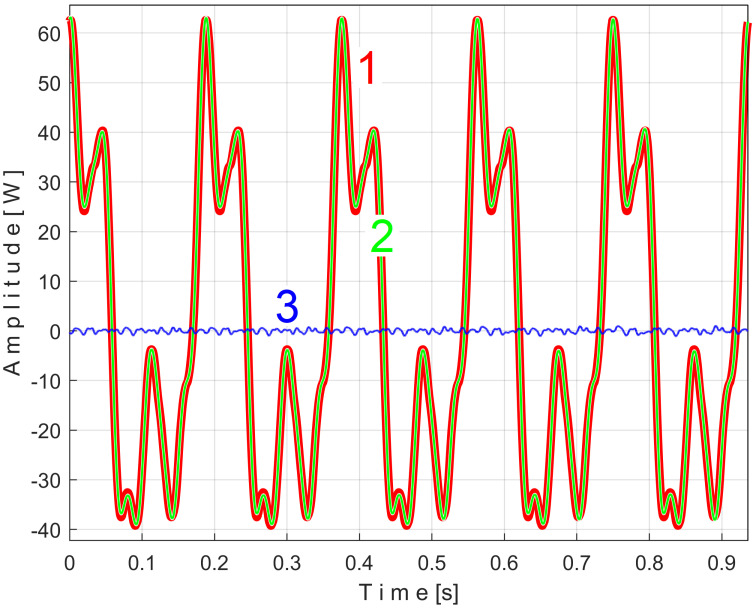
1—The extended pattern *s_PTAe5b_* (*m* = 530, *f_PAb_* = 5.3414 Hz); 2—The analytical description of this pattern (with eight sinusoidal components, Table 2); 3—The residual (the difference between curves 1 and 2).

**Figure 13 sensors-25-01119-f013:**
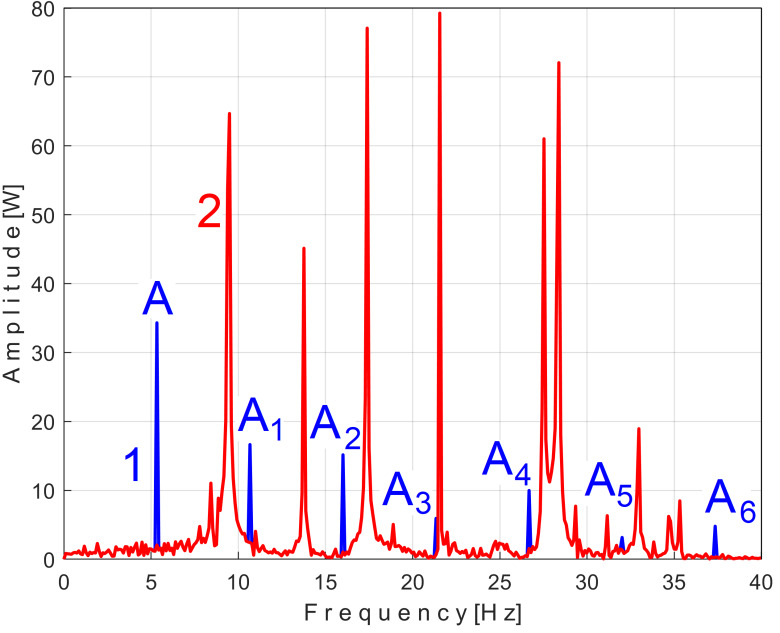
The FFT spectra of: 1—the sequence *s_P50_*; 2—the signal *s_P_*_1*r50*_.

**Figure 14 sensors-25-01119-f014:**
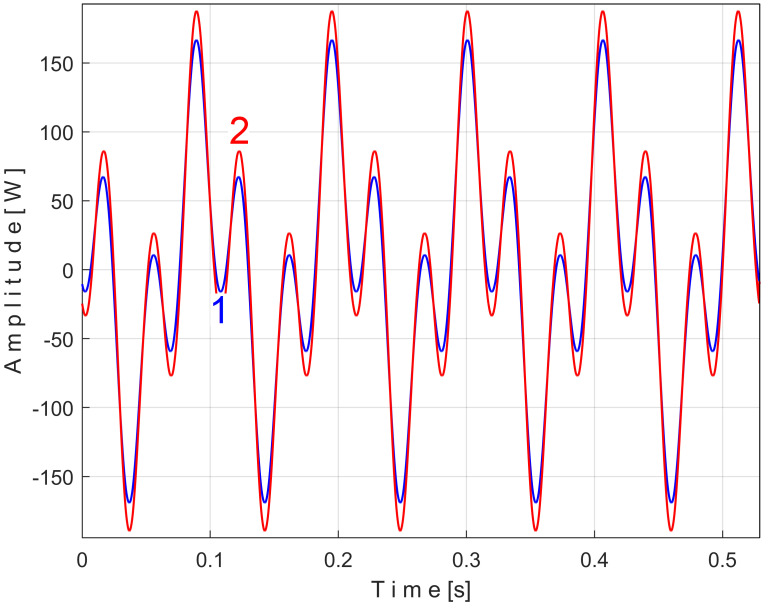
The extended patterns (*m* = 935) describing the behavior of second flat belt in signal *s_P_*: 1—*s_PTBe5a_* (*f_PBa_* = 9.45223 Hz); 2—*s_PTBe5b_* (*f_PBb_* = 9.45981 Hz).

**Figure 15 sensors-25-01119-f015:**
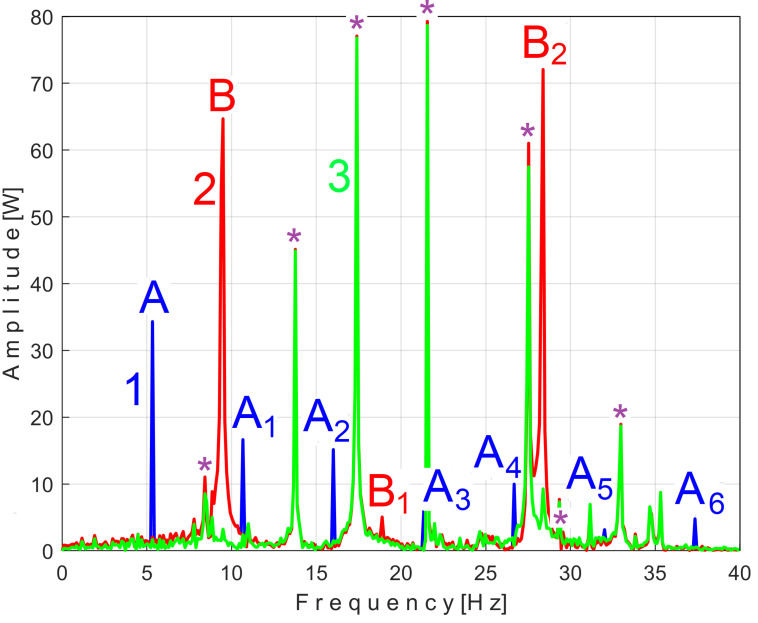
The results of removing the extended patterns *s_PTAe50_* and *s_PTBe89*_* from *s_P50_* in FFT spectrum. 1—The FFT spectrum of the sequence *s_P__50_* (already depicted in Figure 13); 2—The overlapped FFT spectrum of sequence *s_P_*_1*r50*_ (already depicted in Figure 13); 3—The overlapped FFT spectrum of sequence *s_P_*_2*r50*_. The symbol * indicates an abnormal decrease of peaks in spectrum 3.

**Figure 16 sensors-25-01119-f016:**
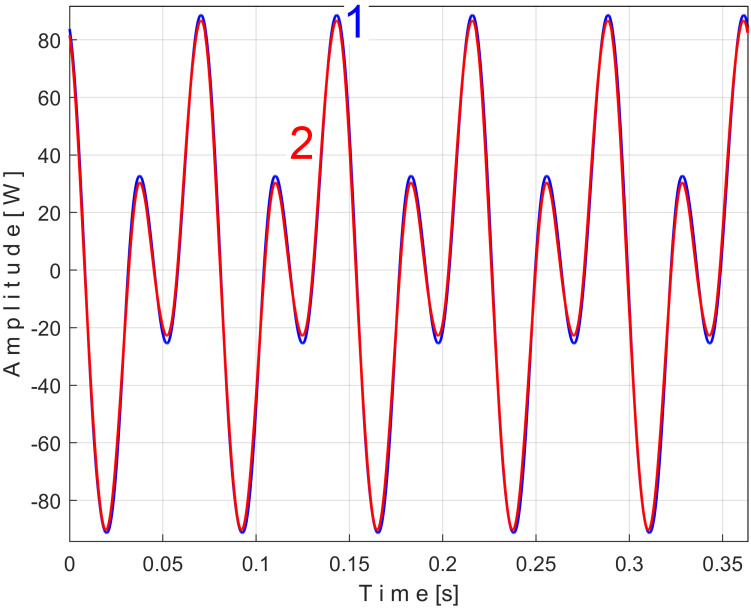
The extended patterns (*m* = 1350) describing the behavior of shaft II in signal *s_P_*: 1—*s_PTCe5a_* (*f_PCa_* = 13.75435 Hz); 2—*s_PTCe5b_* (*f_PCb_* = 13.759 Hz).

**Figure 17 sensors-25-01119-f017:**
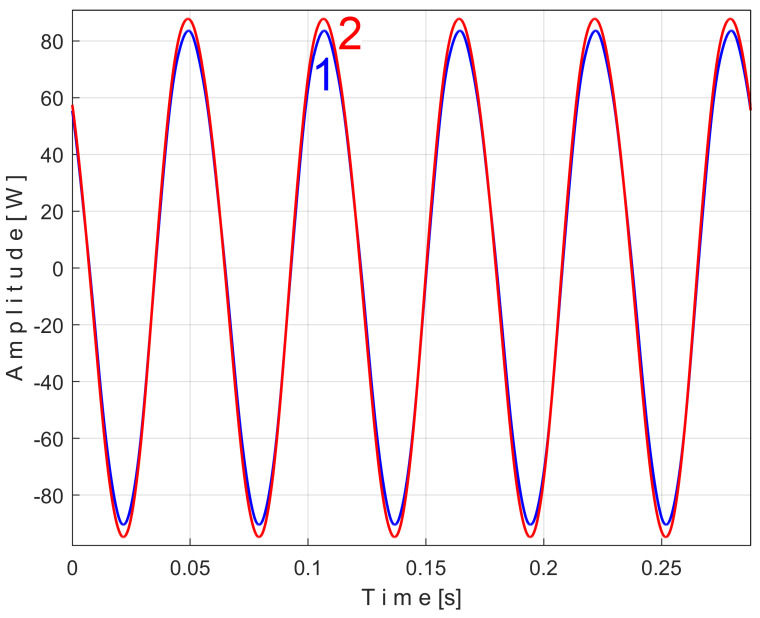
The extended patterns (*m* = 1720) describing the behavior of the shaft III and spindle in signal *s_P_*: 1—*s_PTDe5a_* (*f_PDa_* = 17.37406 Hz); 2—*s_PTDe5b_* (*f_PDb_* = 17.38745 Hz).

**Figure 18 sensors-25-01119-f018:**
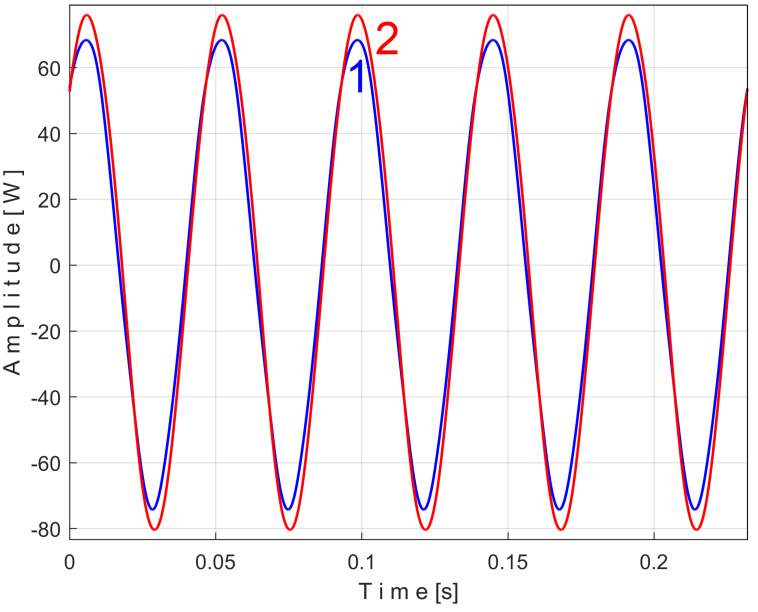
The extended patterns (*m* = 2130) describe the behavior of shaft II in signal *s_P_*: 1—*s_PTEe5a_* (*f_PEa_* = 21.5606 Hz); 2—*s_PTDe5b_* (*f_PEb_* = 21.5781 Hz).

**Figure 19 sensors-25-01119-f019:**
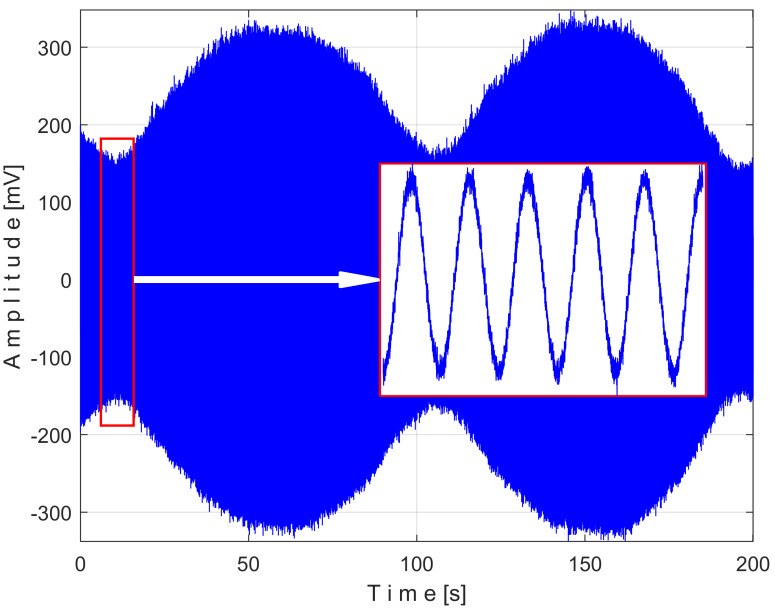
The time-domain representation of the signal V_s_.

**Figure 20 sensors-25-01119-f020:**
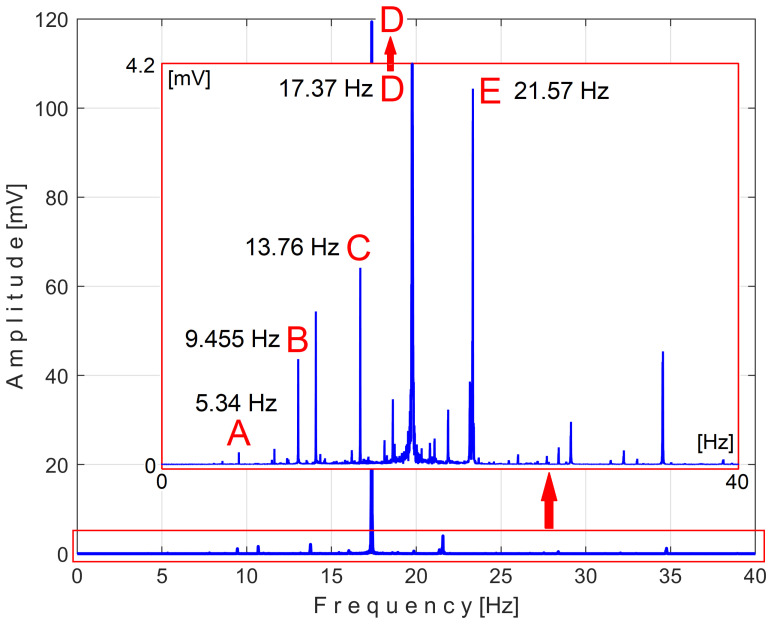
The FFT spectrum of vs. with A, B, …, E the fundamentals of PVSC generated by these MCs: A—the flat belt 1; B—the flat belt 2; C—the shaft II; D—the shaft III and the spindle; E—the shaft I.

**Figure 21 sensors-25-01119-f021:**
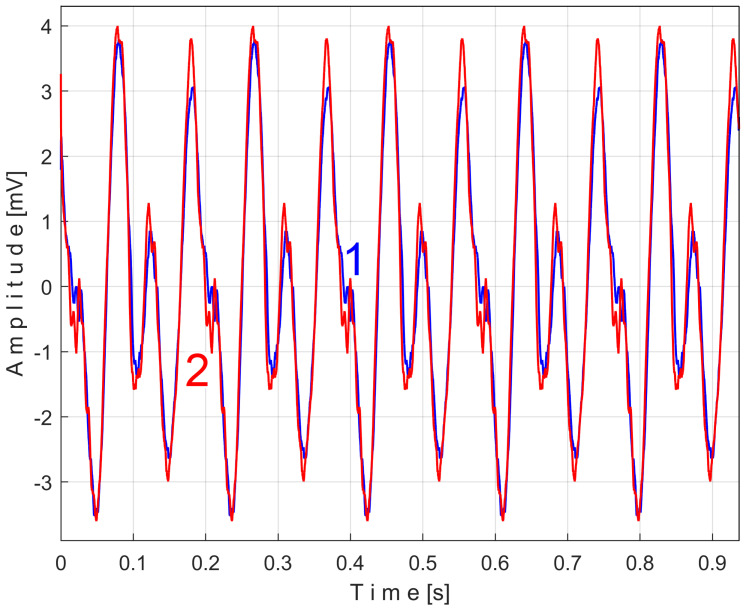
The low-pass filtered extended patterns (*m* = 533) describing the behavior of the flat belt 1 in signal *s_V_*: 1—*s_VTAe5a_*, *f_VAa_* = 5.33736 Hz; 2—*s_VTAe5b_*, *f_VAb_* = 5.34151 Hz.

**Figure 22 sensors-25-01119-f022:**
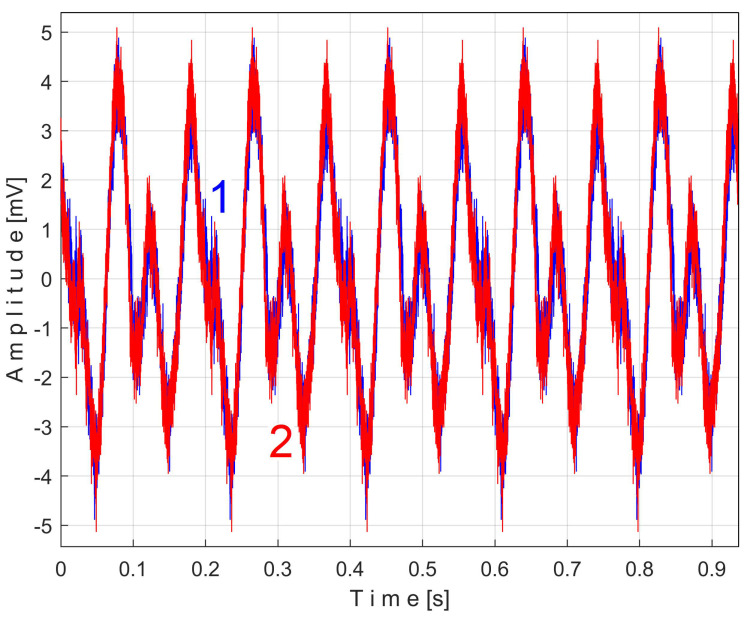
The unfiltered extended patterns 1 and 2 from Figure 21.

**Figure 23 sensors-25-01119-f023:**
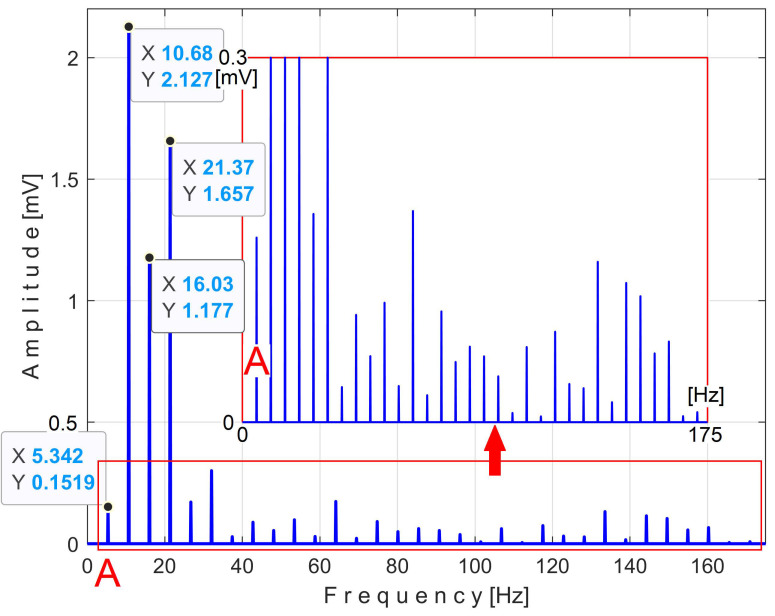
The FFT spectrum of the extended unfiltered pattern *s_VTAe50b_*.

**Figure 24 sensors-25-01119-f024:**
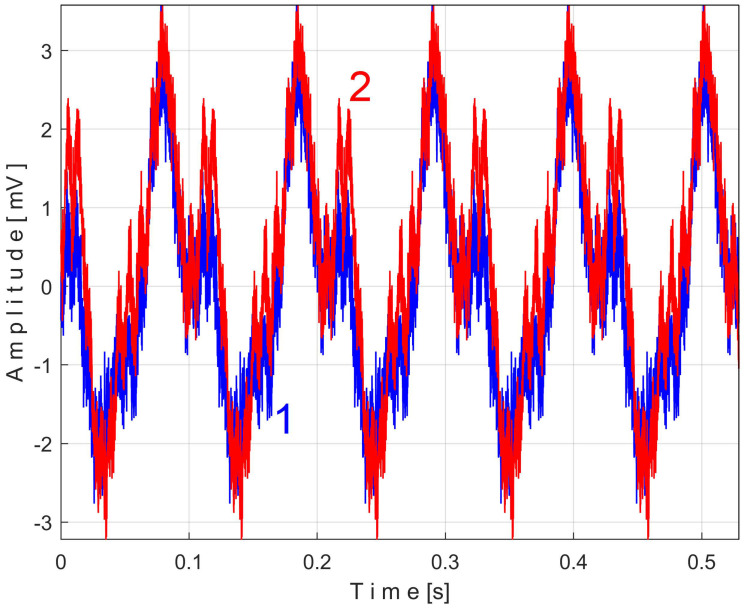
The unfiltered extended patterns (*m* = 935) describing the behavior of the flat belt 2 in signal *s_V_*: 1—*s_VTBe5a_*, *f_VBa_* = 9.4525 Hz; 2—*s_VTBe5b_*, *f_VBb_* = 9.4596 Hz.

**Figure 25 sensors-25-01119-f025:**
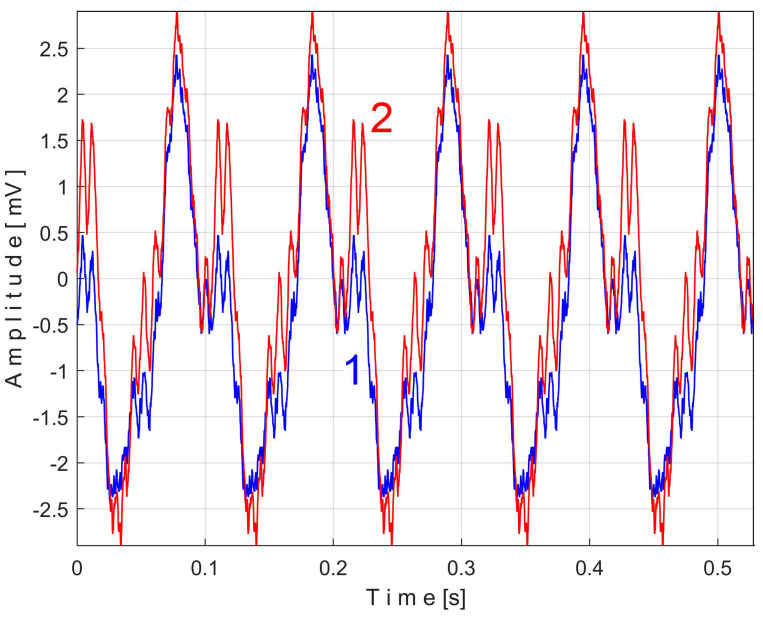
The low-pass filtered extended patterns 1 and 2 from Figure 24 (using a moving average filter with 30 samples in the average).

**Figure 26 sensors-25-01119-f026:**
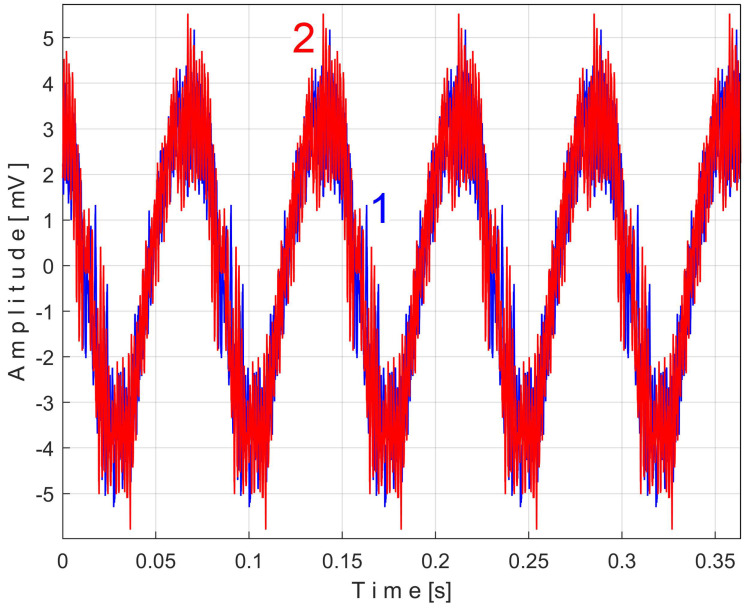
The unfiltered extended patterns (*m* = 1365) describing the behavior of shaft II in signal *s_V_*: 1—*s_VTCe5a_*, *f_VCa_* = 13.75399 Hz; 2—*s_VTCe5b_*, *f_VCb_* = 13.76565 Hz.

**Figure 27 sensors-25-01119-f027:**
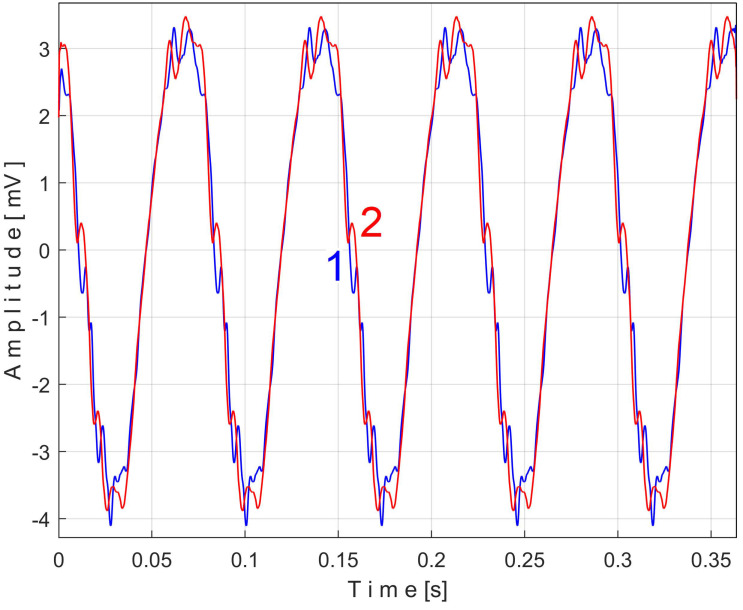
The low-pass filtered extended patterns 1 and 2 from Figure 26 (using a double moving average filter with 37 and 50 samples in the average).

**Figure 28 sensors-25-01119-f028:**
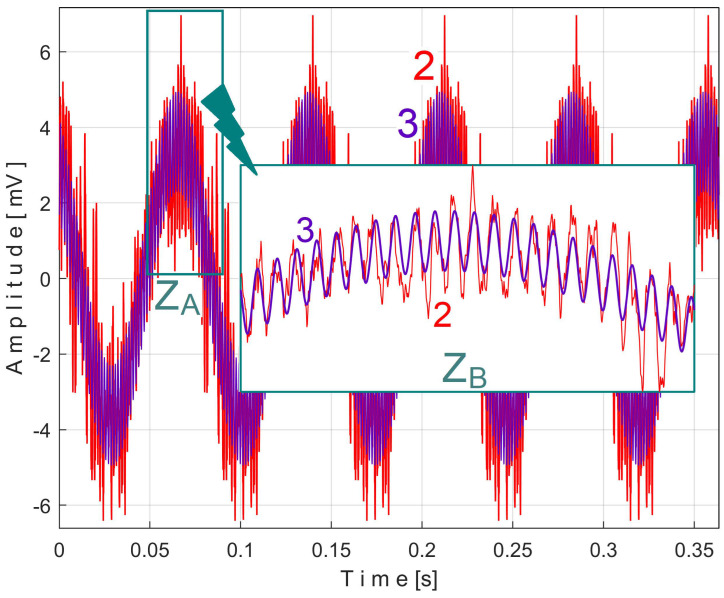
2—The unfiltered pattern *s_VTCe5b_*; 3—An approximation of this pattern based on Equation (7) as *s_VTCe5b_*^2*c*^.

**Figure 29 sensors-25-01119-f029:**
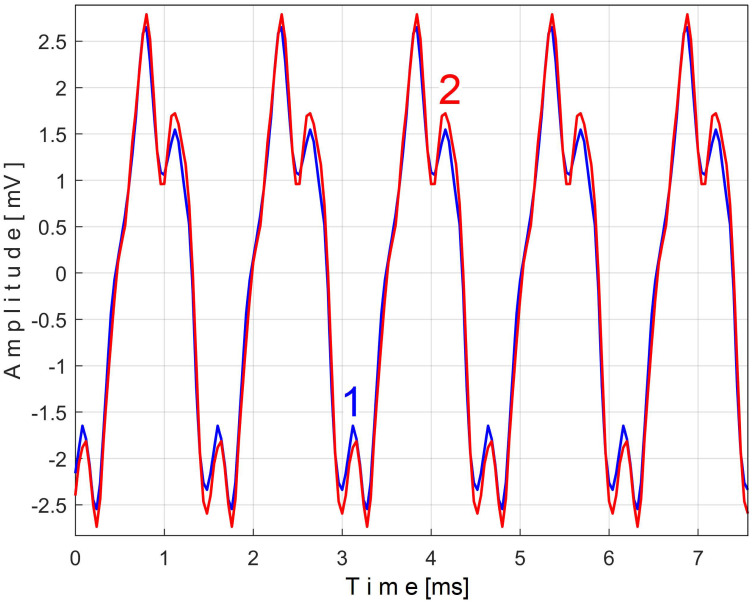
The extended patterns (*m* = 3250) describing the behavior of component *T_VC_*^48^ in signal *s_V_*: 1—*s_VTC_*^48^*_e5a_*, *f_VC_*^48^*_a_* = 660.045986 Hz; 2—*s_VTC_*^48^*_e5b_*, *f_VC_*^48^*_b_* = 660.53362 Hz.

**Figure 30 sensors-25-01119-f030:**
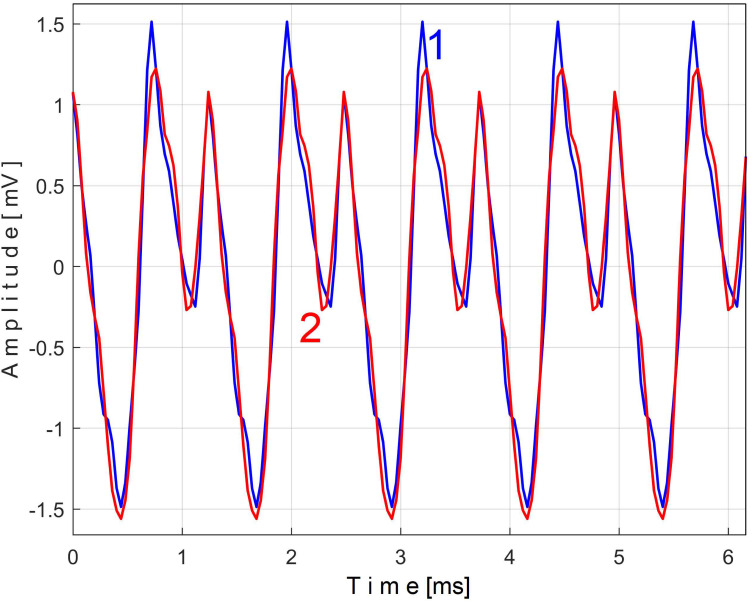
The extended patterns (*m* = 3250) describing the behavior of component *T_VC_*^58^ in signal *s_V_*: 1—*s_VTC_*^58^*_e5a_*, *f_VC_*^58^*_a_* = 797.5646 Hz; 2—*s_VTC_*^58^*_e5b_*, *f_VC_*^58^*_b_* = 798.1551 Hz.

**Figure 31 sensors-25-01119-f031:**
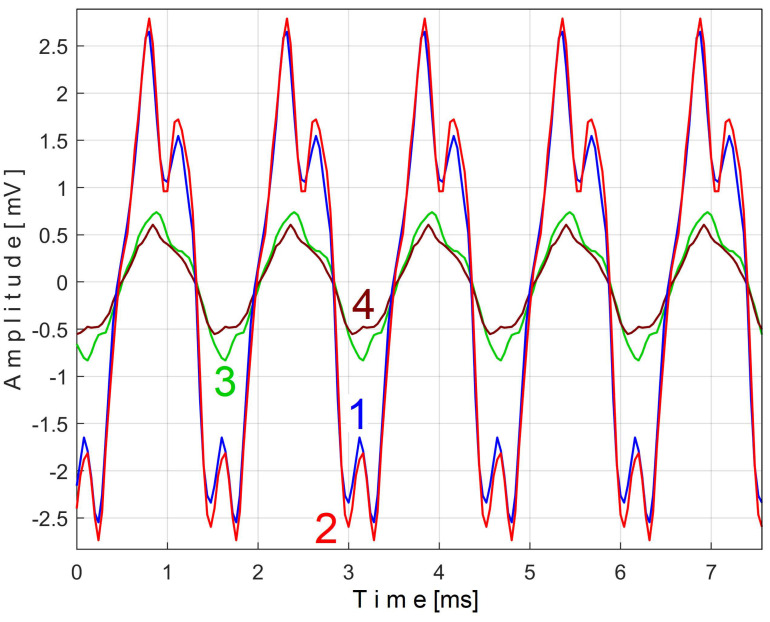
1, 2—The extended patterns (*m* = 3250) with *T_VC_*^48^ in signal *s_V_*: 1—*s_VTC_*^48^*_e5a_*, *f_VC_*^48^*_a_* = 660.045986 Hz; 2—*s_VTC_*^48^*_e5b_*, *f_VC_*^48^*_b_* = 660.53362 Hz. 3,4—The extended patterns (*m* = 65,700) with *T_VC_*^48^ in signal *s_V_*: 3—*s_VTC_*^48^*_e5a_*, *f_VC_*^48^*_a_* = 660.19726 Hz; 4—*s_VTC_*^48^*_e5b_*, *f_VC_*^48^*_a_* = 660.58252 Hz.

**Figure 32 sensors-25-01119-f032:**
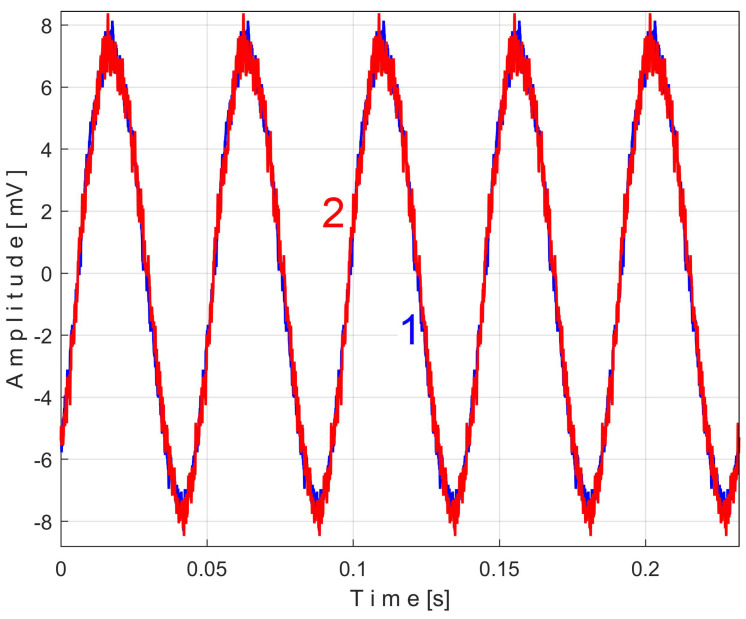
The unfiltered extended patterns (*m* = 2155), which describes the behavior of shaft I in signal *s_V_*: 1—*s_VTCe5a_*, *f_VDa_* = 21.56087 Hz; 2—*s_VTDe5b_*, *f_VDb_* = 21.57886 Hz.

**Figure 33 sensors-25-01119-f033:**
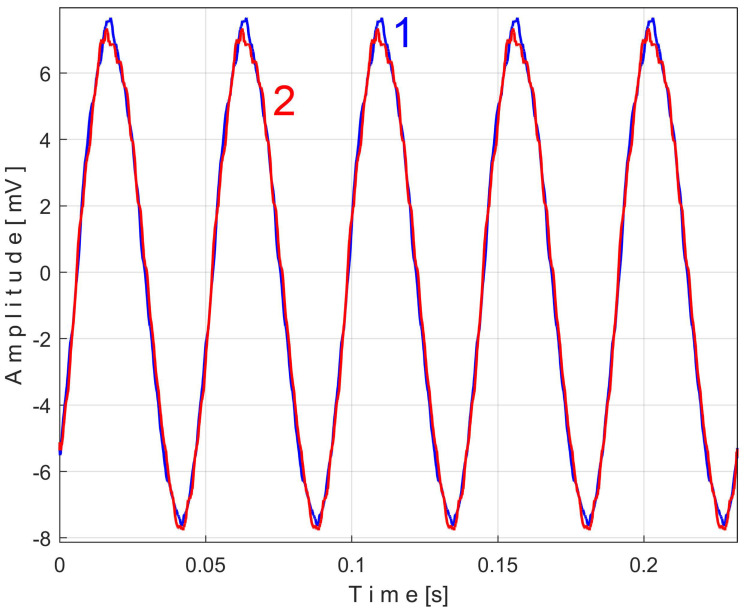
The low-pass filtered extended patterns 1 and 2 from Figure 32.

**Figure 34 sensors-25-01119-f034:**
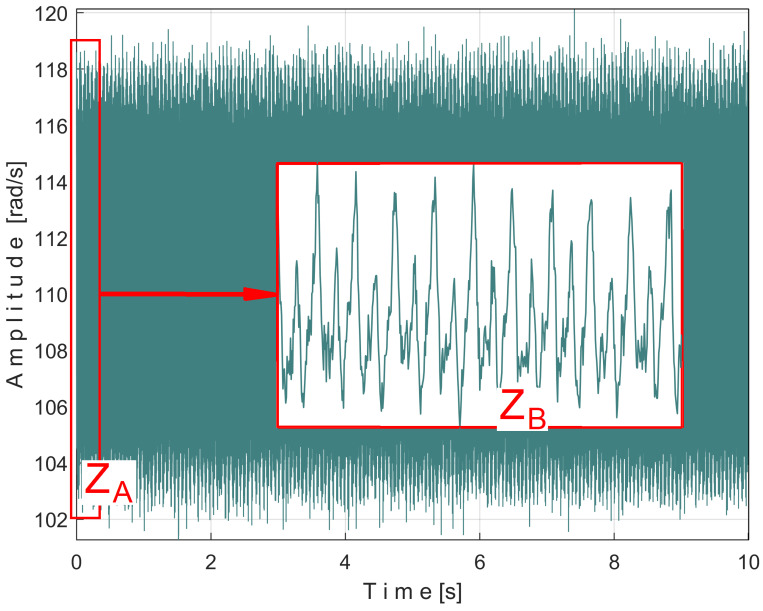
The time-domain representation of the IAS spindle during 10 s of steady-state regime.

**Figure 35 sensors-25-01119-f035:**
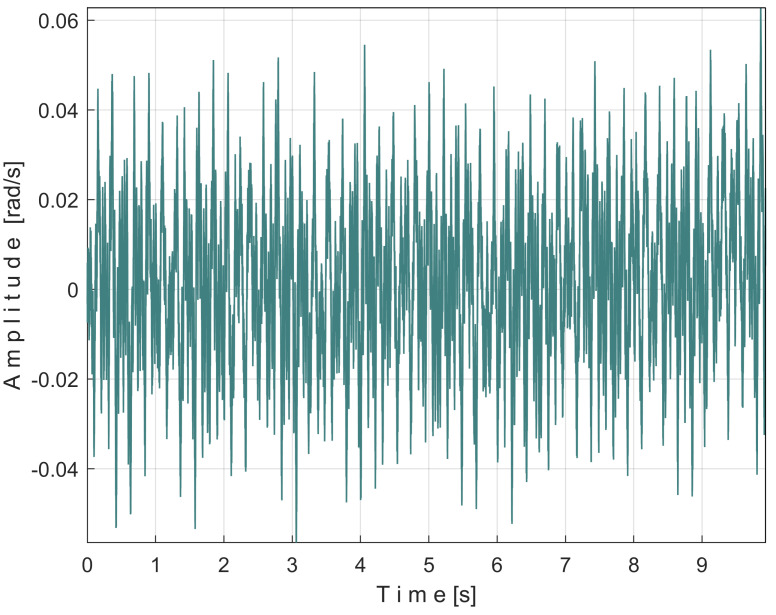
The time-domain representation of variable part of the filtered IAS (*s_I_*) from Figure 34.

**Figure 36 sensors-25-01119-f036:**
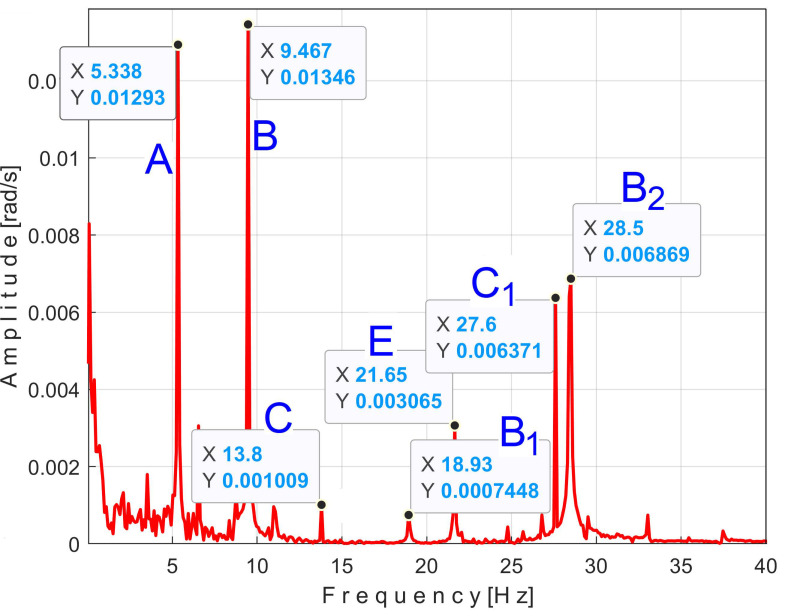
The partial FFT spectrum of signal *s_I_* (0–40 Hz range).

**Figure 37 sensors-25-01119-f037:**
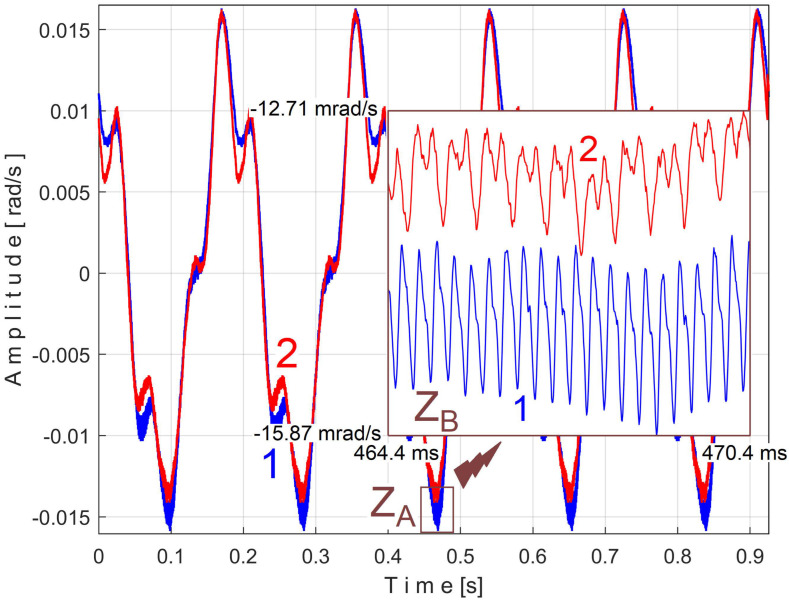
The unfiltered extended patterns (*m* = 25) describing the behavior of the first flat belt in signal *s_I_*: 1—*s_ITAe5a_*, *f_IAa_* = 5.40021 Hz; 2—*s_ITAe5b_*, *f_IAb_* = 5.4107 Hz.

**Figure 38 sensors-25-01119-f038:**
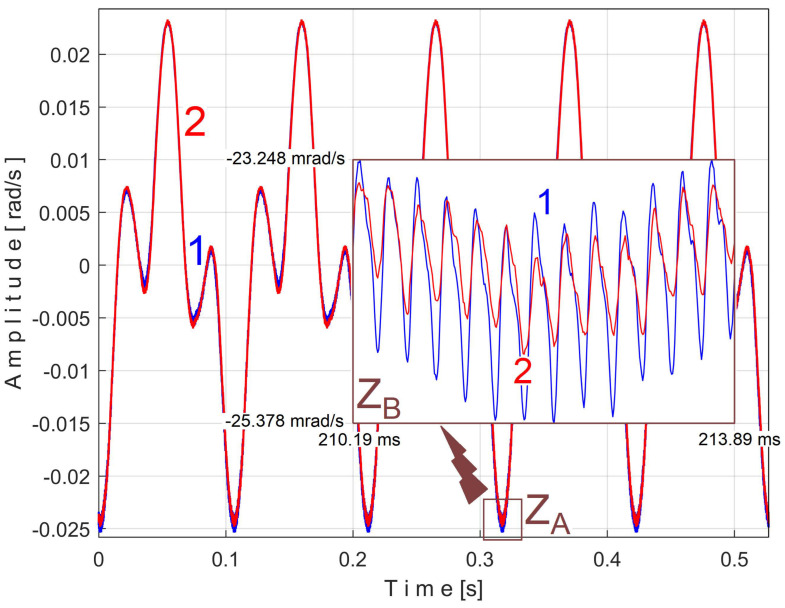
The unfiltered extended patterns (*m* = 46) describing the behavior of the second flat belt in signal *s_I_*: 1—*s_ITBe5a_*, *f_IBa_* = 9.48999 Hz; 2—*s_ITBe5b_*, *f_IBb_* = 9.49061 Hz.

**Figure 39 sensors-25-01119-f039:**
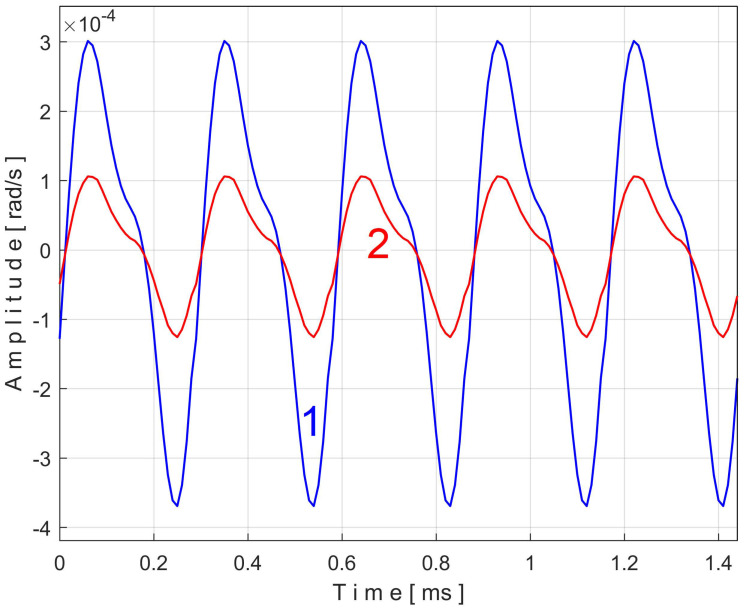
The unfiltered extended patterns (*m* = 17,100) of the false variable component with periods *T_IBa_*^369^ and *T_IBb_*^369^ in signal *s_I_*: 1—*s_ITB_*^369^*_e5a_*, *f_IB_*^369^*_a_* = 3501.7959 Hz; 2—*s_ITB_*^369^*_e5b_*, *f_IB_*^369^*_b_* = 3501.7417 Hz.

**Figure 40 sensors-25-01119-f040:**
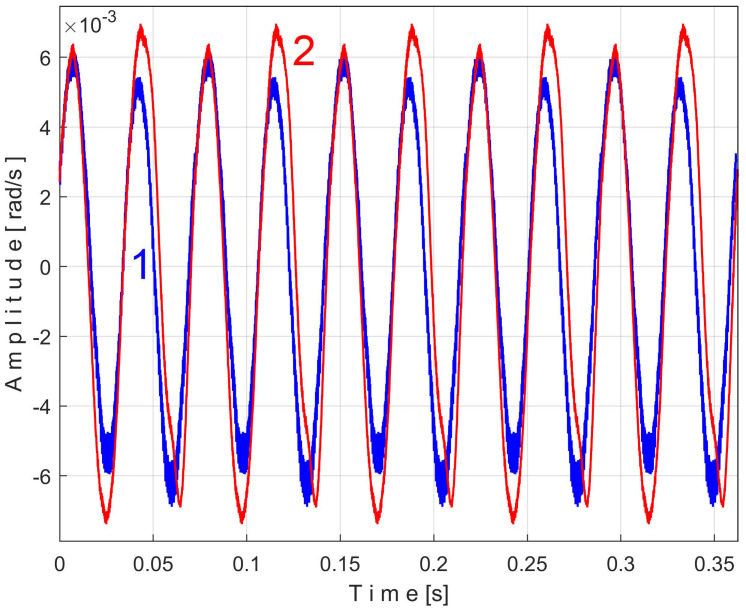
The unfiltered extended patterns (*m* = 68) describing the behavior of the shaft I in signal *s_I_*: 1—*s_ITCe5a_*, *f_ICa_* = 13.826942 Hz; 2—*s_ITCe5b_*, *f_ICb_* = 13.79066 Hz.

**Figure 41 sensors-25-01119-f041:**
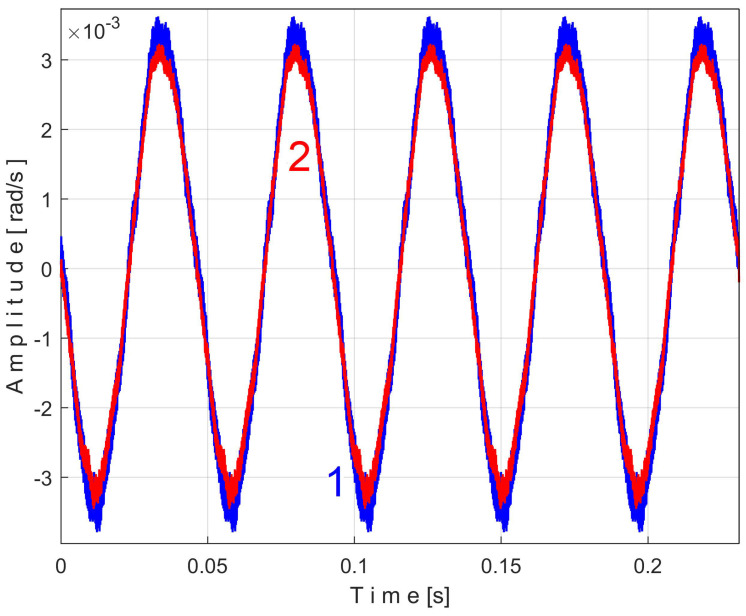
The unfiltered extended patterns (*m* = 106) describing the behavior of the shaft II in signal *s_I_*: 1—*s_ITEe5a_*, *f_IEa_* = 21.6422 Hz; 2—*s_ITEe5b_*, *f_IEb_* = 21.64315 Hz.

**Figure 42 sensors-25-01119-f042:**
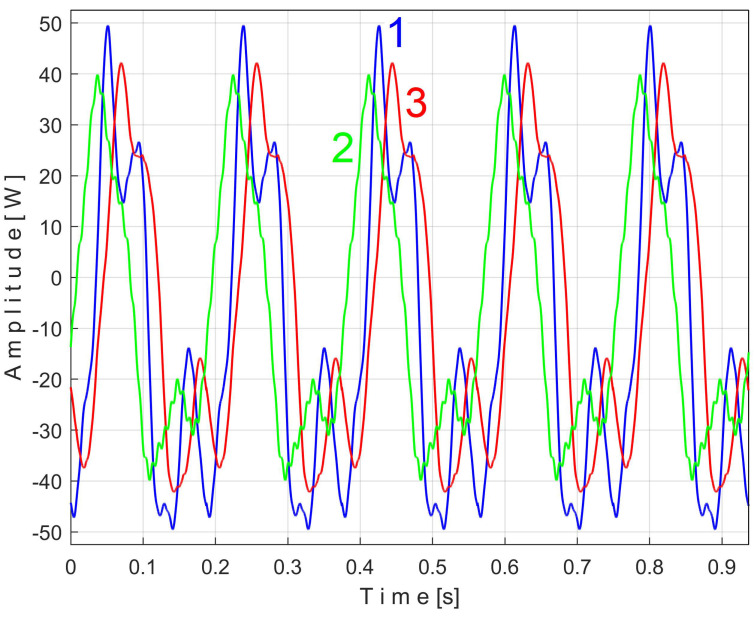
The influence of frequency (period) variation on extended pattern *s_PTAe5a_* shape (from Figure 11, *m* = 530): 1—*s_PTAe5a_* (correct frequency *f_PAa_* = 5.33748 Hz); 2—*s_PTAe5a−_* (deliberately wrong, very slightly lower frequency *f_PAa-_* = 5.335 Hz = *f_Paa_* − 0.00248 Hz); 3—*s_PTAe5a+_* (deliberately wrong, very slightly higher frequency *f_PAa+_* = 5.339 Hz = *f_PAa_ +* 0.00152 Hz).

**Figure 43 sensors-25-01119-f043:**
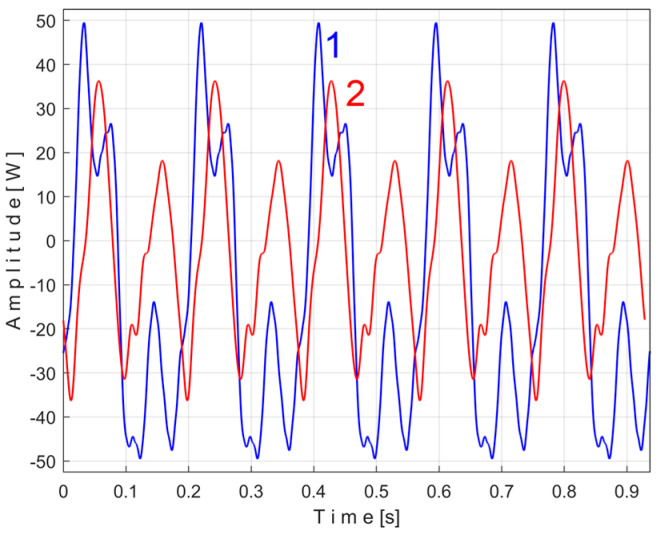
The extended patterns *s_PTAe5a_* (*m* = 530): 1—with all the gearbox parts in rotary motion (a copy of curve 1 from Figure 11, *f_PAa_* = 5.33748 Hz); 2—during the new steady-state regime *f_PAa_* = 5.386069 Hz.

**Figure 44 sensors-25-01119-f044:**
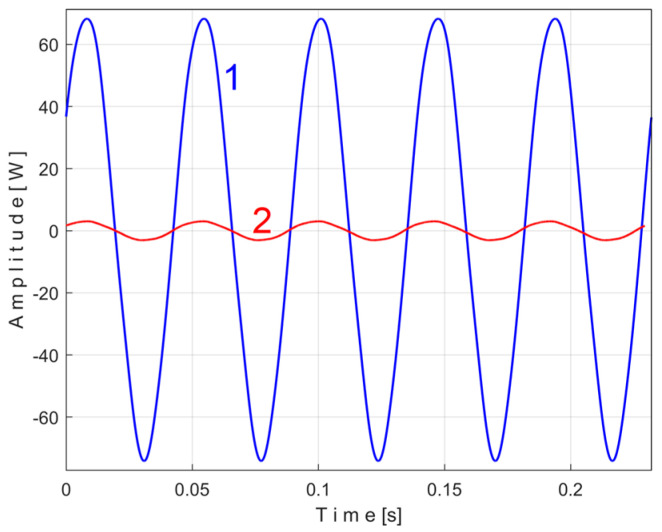
The extended patterns *s_PTEe5a_* (*m* = 2130): 1—with all the gearbox parts in rotary motion (a copy of curve 1 from Figure 18, *f_PEa_* = 21.5606 Hz); 2—during the new steady-state regime, *f_PEa_* = 21.7923 Hz.

**Figure 45 sensors-25-01119-f045:**
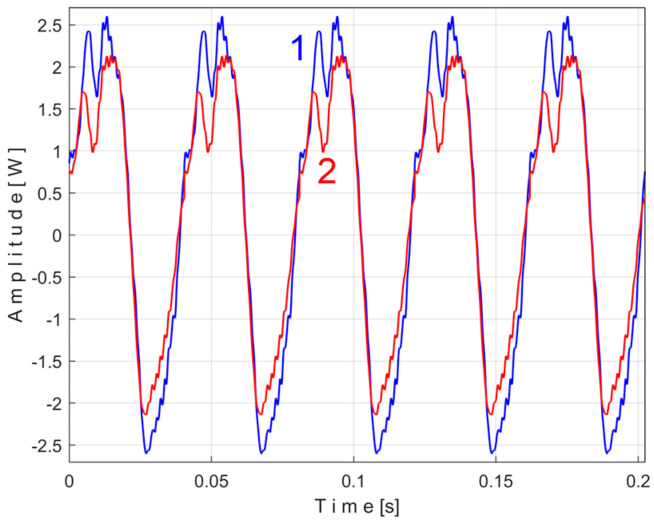
The extended patterns *s_PTMe5_* generated by electric motor (*m* = 2465) during first steady-state regime: 1—*s_PTMe5a_*, *f_PMa_* = 24.69481 Hz; 2—*s_PTMe5b_*, *f_PMb_* = 24.712903 Hz.

**Figure 46 sensors-25-01119-f046:**
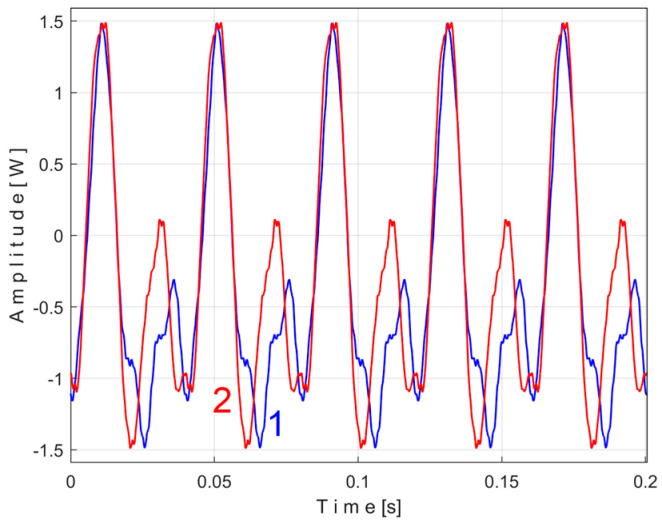
The extended patterns *s_PTMe5_* generated by electric motor (*m* = 2465) during the new steady-state regime: 1—*s_PTMe5a_*, *f_PMa_* = 24.91988 Hz; 2—*s_PTMe5b_*, *f_PMb_* = 24.93528 Hz.

**Figure 47 sensors-25-01119-f047:**
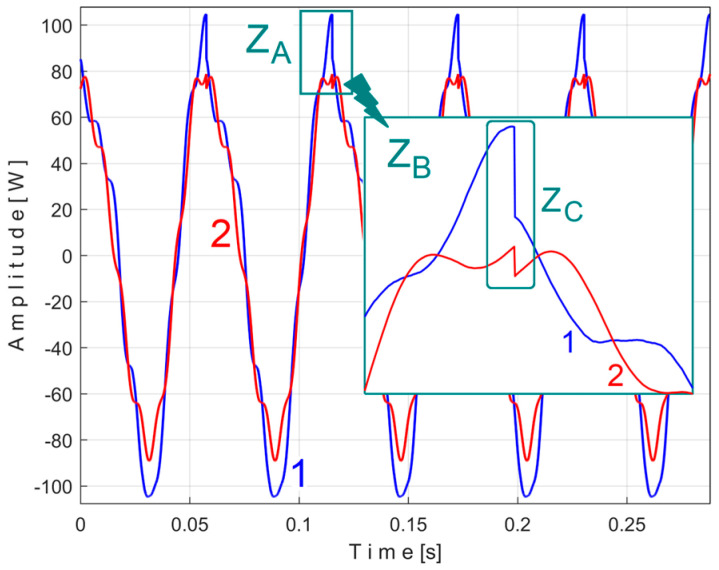
The extended patterns describing the behavior of the shaft III and spindle in signal *s_P_* (using small values of *m*) are 1—*s_PTDe5a_*_1_ (*m* = 20, *f_PDa_*_1_ = 17.35793 Hz) and 2—*s_PTDe5a_*_2_ (*m* = 100, *f_PDa_*_2_ = 17.3638 Hz).

**Table 1 sensors-25-01119-t001:** The average values of frequency of peaks (describing the fundamental of PVSC) A, B, … E.

	*f_PA_* [Hz]	*f_PB_* [Hz]	*f_PC_* [Hz]	*f_PD_* [Hz]	*f_PE_* [Hz]
Found using FFT	5.342	9.454	13.76	17.38	21.58
Found using EMASS	5.3408 (*m* = 1061)	9.4589 (*m* = 1872)	13.76513 (*m* = 2728	17.3865 (*m* = 3453)	21.5761 (*m* = 4288)

**Table 2 sensors-25-01119-t002:** The amplitudes, frequencies, and phases at the origin of time for the sinusoidal components involved in the description (by addition) of the PVSC generated by the first flat belt within the active electrical power.

	F_PA_	H_PA1_	H_PA2_	H_PA3_	H_PA4_	H_PA5_	H_PA6_	H_PA8_
Amplitude [W]	37.19	15.33	15.5	6.268	8.482	2.157	4.499	0.446
Frequency [Hz]	5.341	10.684	16.026	21.374	26.706	32.053	37.401	48.080
Phase at origin of time [rad]	0.9581	0.1707	2.359	1.204	0.5218	2.307	1.266	−1.863

## Data Availability

The data presented in this paper are available upon request and addressed to the corresponding author.

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
