# Peer review of "A Signal Pattern Extraction Method Useful for Monitoring the Condition of Actuated Mechanical Systems Operating in Steady State Regimes"

_sensors, 2025, doi:10.3390/s25041119_

Round 1
Reviewer 1 Report
Comments and Suggestions for Authors
A signal extraction method was presented by the authors. However, many problems were noticed from the manuscript, the author should address for further consideration.
1. In the abstract, the authors included a lot of “(e.g. ……)” after terminologies. There are too many contents regarding “background introduction” in the abstract. The reviewer believed one or two sentences are enough for introducing the background.
2. In the introduction, the authors wrote the portion in a similar way, which includes a lot of brackets (e.g., (usually), (its state), (CSC)). It is understandable to use abbreviations for certain terminologies such as MC, PVSC. However, it is really confusing for people to read the article with abundant brackets in the main body.
3. In the introduction, please properly check the correct style of the reference and make them consistent with each other. The authors wrote “The simplest way to obtain this representation is to filter the variable part of the status signal (converted to numeric format) with a tunable numerical multiple narrow band pass filter [15] having the pass frequencies harmonically correlated on 1/T Hz, 2/T Hz, 3/T Hz and …”, please check whether it is correct to use “Hz” in the sentence.
4. The reviewer cannot understand Eq. (3), further clear explanation is required. The abbreviation “TNMNBP” is too long. The authors wrote “Consider a random noise signal rn where each sample is generated as a random number in the interval [-5 ¸ 5].”, the reviewer is not sure what “¸” mean here.
5. The authors applied the filter to active electrical power signal, vibration signal, and instantaneous angular speed signal. Please explained why it is necessary to include them in the comparison.
6. There are over 50 figures in the manuscript. It is suggested to remove or combine some figures, otherwise it is very hard to follow. There are many grammatical and typing errors in the manuscript, it is highly suggested the authors carefully go through the manuscript.
7. The authors mentioned “The analytical description of AMSSRTI patterns obtained by Curve Fitting Tool facilitates the automatic detection of anomalies in the operation of system components.”, further explanation is required. The advantages of the proposed filter compared with existing one is unclear.
Comments on the Quality of English Language
It needs to be improved.
Reviewer 2 Report
Comments and Suggestions for Authors
The article is written in a coherent manner and addresses a topic that is likely to the be of interest to readers of the journal. The research methodology is appropriate and logical and the discussion is serious. Keywords are appropriate and the quality of the figures and tables is good.
Several small suggestions.
My proposal is to move the description of the test bench from the Results section (lines 260-282) to the beginning of Section 2.
It may be useful for the reader to specify the version of the software (in this case, Matlab).Was only the Curve Fitting Tool from Matlab used in addition?
Overall, my opinion is positive, and I would suggest that the authors continue the work.

Reviewer 3 Report
Comments and Suggestions for Authors
The presented article proposes a new approach to monitoring the condition of mechanical systems operating in steady-state mode. The authors describe a method for extracting periodic components from sensor signals based on averaging signal samples taken at equal time intervals that are multiples of the oscillation period. This allows us to isolate and analyze individual components corresponding to the rotating parts of the mechanism. This approach, essentially being a type of digital filtering, allows us to identify even minor deviations in the system operation that may be signs of malfunctions. The relevance of the work lies in the fact that it proposes a simple and effective method for analyzing signals obtained during monitoring of mechanical systems, which is especially important for systems operating in steady-state mode. The importance lies in the ability to detect deviations in a timely manner, which helps prevent serious breakdowns and reduce maintenance costs. The method was tested on power, vibration, and angular velocity signals, demonstrating its versatility and potential for practical application.
The article needs some revision.
1. The abstract should indicate the relevance of the work, as well as present the results obtained in the article in more detail. 2. The choice of specific parameters for analysis (power, vibration, angular velocity) requires more detailed justification. Why these parameters and not others? What is their relative importance for monitoring the state of a mechanical system, especially in the context of steady-state conditions?
3. What is the effect of noise on the accuracy of the results? How does the developed method cope with different types of noise (white noise, 1/f noise, etc.)? The paper did not analyze the robustness of the method to the influence of noise, and did not describe an approach to noise suppression other than increasing the number of averagings.
4. How are the parameters m (number of averagings) and n (period length in samples) chosen? How do they affect the accuracy and speed of the method? It is worth describing in more detail the optimization of these parameters and the effect of their changes on the result of signal processing.
5. The article mentions that the period may not be a multiple of the sampling interval. How does this fact affect the accuracy of the result? The statement that “the description from Eq. (2) can be considered acceptable” should be made more detailed and more convincing. A more detailed quantitative assessment of the error arising from the “non-integer” period would make this statement more accurate.
6. The article uses a simplified model of a periodic signal as a sum of sinusoids. How realistic is this for real signals of mechanical systems, especially under steady-state conditions?
7. The work compares the results obtained from different sensors. What is the reason that vibration is better suited for analysis (as written in the section) than power? It would also be good to provide a more detailed analysis of the causes of the differences.
8. The article poorly shows the possibility of practical application of the analysis results. It is worth showing in more detail the possibility of practical application of the obtained results. Is calibration or adjustment necessary? How often should measurements be taken? How convenient is this method for use in real conditions? Information on the possible practical implementation of the results of the work can also be shown in the conclusions.
Round 2
Reviewer 1 Report
Comments and Suggestions for Authors
The authors addressed my comments. It is acceptable for its current version.
HOWEVER, personally, I do not like to read a paper with over 47 figrues.
Comments on the Quality of English Language
N/A
Reviewer 3 Report
Comments and Suggestions for Authors
In general, the authors have improved the article and taken into account all my comments.